# A limitation on black-box dynamics approaches to Reinforcement Learning

**Brieuc Pinon**                                    *brieuc.pinon@uclouvain.be*
*Department of Mathematical Engineering*
*UCLouvain*

**Raphaël Jungers**                                 *raphaël.jungers@uclouvain.be*
*Department of Mathematical Engineering*
*UCLouvain*

**Jean-Charles Delvenne**                           *jean-charles.delvenne@uclouvain.be*
*Department of Mathematical Engineering*
*UCLouvain*

**Reviewed on OpenReview:** *https://openreview.net/forum?id=wPHVijYksq*

## Abstract

We prove a fundamental limitation on the computational efficiency of a large class of Reinforcement Learning (RL) methods. This limitation applies to model-free RL methods as well as some model-based methods, such as AlphaZero. We provide a formalism that describes this class and present a family of RL problems provably intractable for these methods. Conversely, the problems in the family can be efficiently solved by toy methods. We identify several types of algorithms proposed in the literature that can avoid our limitation, including algorithms that construct an inverse dynamics model, and planning algorithms that leverage an explicit model of the dynamics.

## 1 Introduction

An important research avenue in Artificial Intelligence is dedicated to creating, analyzing, and evaluating Reinforcement Learning (RL) methods (Russell & Norvig, 2010). One goal is to understand when and why some methods perform better than others from a statistical and computational point of view. Explaining such differences is central to guiding the design of new efficient algorithms.

A first step in understanding the differences between methods is to abstract them into classes. Two of the main classes of RL methods are model-based and model-free methods. Model-based methods are algorithms that leverage a known or learned model of the environment dynamics (Mordatch & Hamrick, 2020). In contrast, model-free methods, such as Q-learning and Policy Optimization algorithms (Sutton & Barto, 2018), do not use such a model.

Although the distinction between model-free and model-based classes is commonly accepted (Sutton & Barto, 2018), there are no agreed-upon general formal definitions of these classes. Authors resort to proposing their own definitions (Sun et al., 2019) or to proving their results on classical algorithms representative of these classes (Tu & Recht, 2019).

Several works have studied the relative performance of these classes from a theoretical point of view, and it is cited as an open problem in the survey Levine et al. (2020). For tabular Markov Decision Process, Tu & Recht (2019) review the existing literature that studied the model-free or model-based methods. They find no clear conclusion in favor of one class over another. In the specific problem family of the Linear Quadratic Regulator, Tu & Recht (2019) demonstrate a polynomial separation result in favor of a model-based method.

To our knowledge, Sun et al. (2019) provide the only result with a gap in the efficiency that is exponential in a relevant parameter of the problems. They offer a definition for the class of model-free methods and then present a family of problems intractable for model-free methods but easy for a specific model-based method.

In this work, we define the class of black-box dynamics RL methods containing model-free RL (Q-learning, Policy Optimization, . . . ) and several RL methods considered to be model-based such as AlphaZero (Silver et al., 2017). These methods do not leverage (learned) knowledge of the dynamics or except possibly by generating simulations of transitions. Our paper unravels explicitly a limitation on this class of black-box dynamics methods and reviews potential ideas in the literature to circumvent this limitation.

To establish our result, we formalize the class of black-box dynamics methods in Definition 3.2. The definition aims to achieve two objectives: (1) to capture a large class of methods encompassing model-free and some model-based RL methods, and (2) to allow the proof of a limitation on these methods. Our definition relies on the fact that black-box dynamics methods can be formulated to interact with problems through an *interface*, a novel concept we introduce (pictured in Figure 1). An RL method will fit our definition if the method can still be implemented when our interface is placed between the method and the problem. This interface allows us to formulate and check conditions on the information that flows from the problems to the black-box dynamics methods. In particular, states are not directly fully observable to the method through the interface but only partial information that allows to get (Q-)value and policy functions evaluations.

Next, we introduce a family of problems provably intractable for black-box dynamics methods. However, this family of problems is easy to solve and can be provably efficiently solved by a toy method that identifies a rewarding state and learns to reach it. Moreover, we perform numerical experiments and show that a planning algorithm leveraging a learned model of the dynamics also efficiently solves the problems in the family.

Our work differs from the result of Sun et al. (2019) in two ways. First, we broaden the class of methods with a limitation to some model-based methods such as AlphaZero. We obtain this result by leveraging our concept of interface to formalize this larger class of methods.

Our second contribution with respect to the result of Sun et al. (2019) is that our family of problems can be efficiently solved without strong specific priors about it. In contrast, in their result, the success of the model-based method crucially relies on an a priori known state to reach. They encode this knowledge into the model-based method to its advantage. In general, such knowledge cannot be assumed to be known in practice and impairs the practical applicability and generality of their claim.

In summary, our work identifies a failure-case for a broad range of general-purpose RL methods. Moreover, we review potential solutions to this flaw in the literature.

This article is structured as follows. Section 2 defines the notation and formalizes the problems we address. Section 3 characterizes the black-box dynamics RL methods with our definition and linked interface. Section 4 states our main theoretical result. Section 5 demonstrates numerically the performance of several RL methods. Finally, Section 6 discusses ideas in the literature that could overcome the limitation presented in this paper, and Section 7 provides a summary of our findings.

Appendix B presents an alternative formalization choice with its implications, and Appendix C discusses the differences with the work of Sun et al. (2019) in more detail.

## 2 Preliminaries

**Notation** We use $\Delta(\Omega)$ to denote the set of probability measures over a sample space $\Omega$ with an implicitly associated $\sigma$-algebra. We define the function $\mathbf{1}(.)$ to output 1 if the condition in its argument is respected, else 0. For a set $A$, we note $A^* = \cup_{i \in \mathbb{N}} A^i$ the set of all finite sequences of elements in $A$.

In this paper an *RL problem* is a finite horizon Markov Decision Process (MDP), which is defined by a horizon $H \in \mathbb{N}$, a state space $\mathcal{S}$, an action space $\mathcal{A}$ and an operator $P$, which determines an initial state distribution with $P_0(s_0) \in \Delta(\mathcal{S})$ and a dynamics with $P_{\mathrm{dyn}}(r, s' | s, a) : \mathcal{S} \times \mathcal{A} \to \Delta(\mathbb{R} \times \mathcal{S})$, where $r$ is the reward and $s'$ is the next state.

Throughout the paper, the rewards are in the interval $[0, 1]$, the set of actions is binary $\mathcal{A} = \{0, 1\}$, and the sets of states $\mathcal{S}$ are of the form $\{(t, x) \in \{0, \ldots, H\} \times \mathbb{R}^n\}$ for some $n \in \mathbb{N}$, where $t$ is the time step. Initial states have $t = 0$, and $t$ is incremented at each transition by $P_{\text{dyn}}$. When the time step $H$ is reached, we say that the state is final and the trajectory ends. Our formalization is a particular case of the general framework of Contextual Decision Processes (Jiang et al., 2017).

We note $(s_0, a_0, r_0, s_1, a_1, \ldots, s_H) \sim P^\pi$ a trajectory sampled according to the operator $P$ and a policy $\pi : \mathcal{S} \to \Delta(\mathcal{A})$. We use $\pi^U$ to denote the policy which outputs a uniform distribution over actions.

The objective is to find a policy that maximizes the expected cumulative rewards $\mathbb{E}_{(s_0, a_0, r_0, \ldots, s_H) \sim P^\pi}[\sum_{t=0}^{H-1} r_t]$. An *RL method* is an algorithm that outputs a policy given access to $P$. We note that the policy returned by the RL method might not be optimal. The method *solves* the MDP if it outputs an optimal policy.

## 3 Formalization of black-box dynamics RL methods

In this section, we define the class of black-box dynamics RL methods (Definition 3.2) and then state a limitation on their efficiency in the next section.

**The formalism** We defined in the last section an RL method as an algorithm that outputs a policy given $P$ (the operator describing the dynamics of an RL problem), but we did not formally specify how $P$ is accessed by the method. In the literature, it is common to assume that $P$ can be used to sample trajectories from a chosen policy. Here instead, black-box dynamics RL methods access $P$ through an *interface*. An algorithm for RL problems is considered black-box dynamics if it can be implemented in an RL method interacting only with the interface (Definition 3.2).

The interface, defined in Algorithm 2, is a set of algorithms (oracles) that can be called to get information about the RL problem. The interface is thus placed between the RL problem and the RL method as pictured in Figure 1. This setup allows us to formulate and check conditions on the information that can flow from the RL problem to the black-box dynamics RL method.

To build intuition on the information that we impose the interface to filter, consider deep Q-learning methods as an example. In these methods, the Bellman equation might be enforced by minimizing an empirical average of $\big[Q(s, a) - (r + \max_{a'} \bar{Q}(s', a'))\big]^2$ for some sampled transitions $(s, a, r, s')$ and where $\bar{Q}$ is an old estimate of the Q-values. We observe that in the Bellman equation, the state $s'$ is not directly needed but only its evaluation through $\bar{Q}$.

More generally, across a broad range of RL methods relying on the Bellman equation, given a transition $(s, a, r, s')$, the state $s'$ is only used through evaluations of learned ML models such as value or Q-value functions.

With the help of the interface, our Definition 3.2 formulates an abstract condition that these indirect accesses to $s'$ satisfy in black-box dynamics methods. More precisely, when a transition $s, a, r, s'$ is sampled from $P$, the interface does not directly return $s'$ to the calling method, but $\mathcal{F}(s', D)$ where $D$ is a dataset and $\mathcal{F}$ is a function chosen by the calling algorithm. The dataset $D$ is a list of partial information $(s, a, r, f)$ about past sampled transitions $(s, a, r, s')$ where $f$ is the evaluation of $s'$ by $\mathcal{F}$ with an old version of the dataset.

The function $\mathcal{F}$ is chosen by the RL method and can be selected to be a Machine Learning (ML) algorithm to learn and use (Q-)value and/or policy functions. In that case, $D$ can play the role of a replay buffer, with it, $\mathcal{F}(s, D)$ can learn a neural network according to the Bellman equation and evaluate the state $s$ to extract Q-values. This allows the method to choose an action based on estimated Q-values.

We ask $\mathcal{F}$ to respect a constraint on its output without obstructing the implementation of our methods with the interface. Specifically, we demand that $\mathcal{F}$ is invariant to shufflings of the coordinates in the states. This condition is naturally satisfied by ML algorithms since, without a priori, a learning process is expected to treat the features symmetrically. This condition allows us to model the reliance of methods on rewards

to extract information from states. This is leveraged in our Theorem 4.1 to prove that some states are indistinguishable from the point of view of a black-box dynamics RL method.

To implement an RL method with the interface, the RL method should also be able to ask new transitions to generate trajectories or to do local planning by testing several possible actions from a given state. To permit these with the interface without revealing the state $s'$, the interface returns a number $\bar{s}'$ that the RL method can use to ask for new transitions. In particular, we choose $\bar{s}'$ to be equal to the iteration number at which the state is sampled since the start. For example, at the start, the RL method can ask a new initial state; the interface numbers that state $0$ and provide this number to the RL method; the RL method can ask for new transitions to the interface that leads to successive states numbered $1, 2, \ldots, 17$; at that point, the RL method could ask the interface to sample a new transition from the $12$th state with action $a = 1$; the interface then responds to the method with the obtained reward, $18 (= \bar{s}')$, and $\mathcal{F}(s', D)$ where the dataset $D$ contains information about the $17$th first sampled transitions. Simultaneously, the dataset is augmented with the transition from the $12$th to the $18$th state for future (improved) state evaluations with $\mathcal{F}$.

In some algorithms, transitions are simulated with a learned model. It is not generally possible to learn a model of the dynamics with the interface under our Definition 3.2. However, it is possible to perfectly simulate the dynamics using calls to the interface that interacts directly with $P$ (describing the true underlying dynamics).

**Black-box dynamics RL methods**  Our Definition 3.2 below encompasses a large set of classical RL methods that includes model-free RL approaches such as Policy Optimization and Q-learning. The definition also includes some model-based RL methods that only use a model of the dynamics to generate new transitions such as AlphaZero (Silver et al., 2017). These methods treat a model of the dynamics as an input-output black-box function and do not leverage its internal computations.

On the other hand, there are several examples of algorithmic schemes that our definition does not allow. We defer to Section 6 a discussion on RL methods from the literature that are excluded from our definition but describe shortly here some examples.

Learning a model of the dynamics is not allowed, except if it is only used to sample transitions since in that case these transitions can directly be generated with the interface. Thus, the definition does not allow taking a gradient through a learned model or looking inside a learned model of the dynamics.

Also, learning a model of the inverse dynamics is not allowed, such as learning to predict the action to take given the current state and a wanted future state.

**Definitions**  As explained above, a black-box dynamics RL method interacts with an interface linked to an RL problem, Algorithm 2 pictured in Figure 1. The interface is a set of algorithms with an internal state maintained between calls made by the RL method. Through the interface, the RL method can ask for new transitions, the interface then returns two objects for any sampled state $s'$. The first object is a number $\bar{s}'$ computed by an encoder-decoder described below. The interface also returns a real vector $f$: the evaluation of a constrained function $\mathcal{F}$ on $s'$ and an internal dataset $D$. This second mode of state access allows the RL methods to train and evaluate ML models on states, such as learning value functions using neural networks.

At each call for a new transition, the interface computes $\bar{s}'$ and $f = \mathcal{F}(s', D)$ then returns these values to the RL method. Simultaneously, the interface builds incrementally its internal dataset $D$ with this information. Similarly to RL methods when ML algorithms are used, the ML models are trained from the data, and the data is generated with the help of the ML models. This cyclical dependency is represented by the curved arrows in Figure 1.

The state reference $\bar{s}'$ is computed by an encoder-decoder of states defined in Algorithm 1. The encoder and decoder translate the states to numbers, and numbers to states respectively. The algorithm numbers the encountered states in the order of their visit (potentially associating several numbers to one state if it is visited several times). This numbering ensures that no information about the state is revealed to the calling method. We use $\bar{s}$ to denote a computed number corresponding to some state $s$.

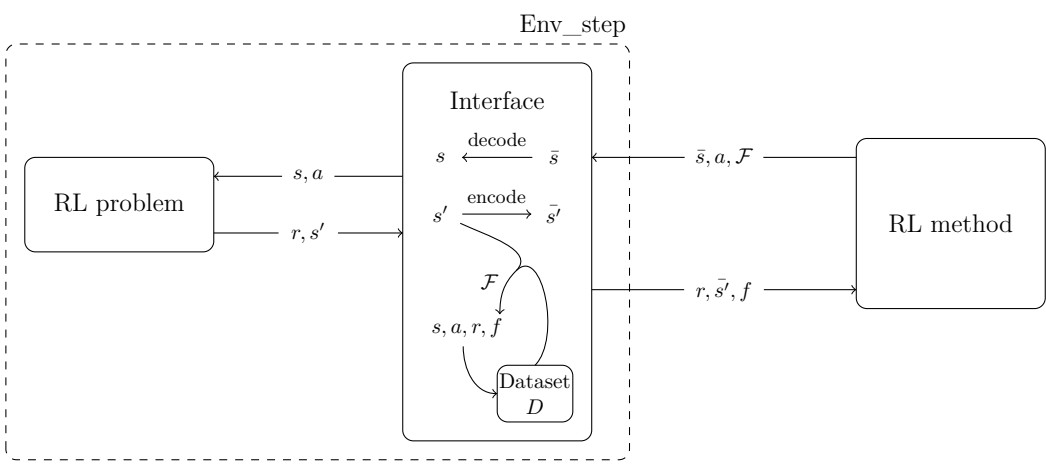

Figure 1: Representation of an RL method that handles the interface linked to an RL problem. The interface allows filtering the information that can flow from the RL problem to the RL method by design. In this Figure, the Env_step algorithm of the interface is pictured. This algorithm allows the RL algorithm to draw a transition from the RL problem. The encountered state $s'$ is passed to the RL algorithm through a reference number to the state, $\bar{s}'$, computed by an encoder-decoder defined in Algorithm 1. The state is also passed to the method through a function evaluation, $f = \mathcal{F}(s', D)$, defined by a dataset $D$, and a chosen function $\mathcal{F}$. This dataset is internal to the interface and incrementally constructed from previous transitions. The function $\mathcal{F}$ can be chosen to implement a learning algorithm that uses the dataset to train a Machine Learning model and evaluate it on the generated state $s'$, the result can be used as a value function for example.

---

**Algorithm 1** Encoder and decoder of states.

---

    last_idx ← -1
    list ←[]
    **function** ENCODE_STATE($s$)
        list ←list.push($s$)
        $\bar{s}$ ←last_idx+1
        **output** $\bar{s}$
    **function** DECODE_POINTER($\bar{s}$)
        **output** $s = \text{list}[\bar{s}]$

---

**Algorithm 2** Env : an interface linked to an RL problem defined by the transition operator $P$. The interface is a set of algorithms with a common internal state. The interface has access to an initialized encoder-decoder of states as defined in Algorithm 1.

---

    $D ← []$                                             ▷ internal dataset
    **function** Env_init($\mathcal{F}$)
        $s_0 \sim P_0$
        $\bar{s}_0 ← \text{encode\_state}(s_0)$
        **output** $\bar{s}_0, \mathcal{F}(s_0, D)$
    **function** Env_step($\bar{s}$, $a$, $\mathcal{F}$, append_to_data = True)
        $s ← \text{decode\_pointer}(\bar{s})$
        **if** $s$ is not final **then**
            $r, s' \sim P_{\text{dyn}}(r, s' \,|\, s, a)$
            $\bar{s}' ← \text{encode\_state}(s')$              ▷ number linked to the state revealing no information
            $f ← \mathcal{F}(s', D)$         ▷ allows to train and evaluate (Q-)value and policy functions
            **if** append_to_data **then**          ▷ set to False to not add the transition to the dataset
                $D ← D.\text{append}((s, a, r, f))$
            **output** $r, \bar{s}', f$
        **else if** $s$ is terminal **then**
            **output** $\perp$
    **function** Env_eval_state($s$, $\mathcal{F}$)
        **output** $\mathcal{F}(s, D)$
    **function** Env_encode($s$)
        **output** encode_state($s$)

---

Our Definition 3.2 requires that the function $\mathcal{F}$ respects a special symmetry condition under permutations of the input coordinates. We define such permutations here in Definiton 3.1.

**Definition 3.1.** A *permutation of coordinates* $p : \mathbb{R}^n \to \mathbb{R}^n$ is a function such that for $x \in \mathbb{R}^n$, $p(x)_{u(i)} = x_i$ for some bijective function $u$ from $\{1 \dots n\}$ to itself.

We also use interchangeably $p' : \mathbb{N} \times \mathbb{R}^n \to \mathbb{N} \times \mathbb{R}^n$ defined as $p'((t, x)) = (t, p(x))$.

From these definitions, we can express Definition 3.2 of black-box dynamics RL methods.

**Definition 3.2.** A *black-box dynamics RL method* interacts only with the interface (Algorithm 2) linked to the RL problem. The function $\mathcal{F}$ given in argument to the algorithms of the interface must have the following form $\mathcal{F} : \mathcal{S} \times D \to \mathbb{R}^N$ where $N$ is a natural number and $D \in (\mathcal{S} \times \mathcal{A} \times \mathbb{R} \times \mathbb{R}^N)^*$ is a dataset. Moreover, for any permutation of the coordinates $p$, we must have $\mathcal{F}(s, ((s_0, a_0, r_0, f_0), \dots)) = \mathcal{F}(p(s), ((p(s_0), a_0, r_0, f_0), \dots))$ for any state $s \in \mathcal{S}$ and $(s_0, a_0, r_0, f_0), \dots \in \mathcal{S} \times \mathcal{A} \times \mathbb{R} \times \mathbb{R}^N$.

**Interpretation and examples** This formulation of the Definition 3.2 reaches two objectives:

    1. To allow a large class of common RL methods to fit the definition.

2. To constrain the information that can pass from the RL problem to the RL method, allowing us to prove the limitation in the next section.

The definition reaches the first objective by allowing to encode common patterns in RL methods. We provide several examples of this flexibility.

The first step in converting an RL method to fit the interface is to replace any generation of transition $(s, a, r, s')$ by a call to the interface defined in Algorithm 2, in particular, the Env_step algorithm pictured in Figure 1.

The reward $r$ is directly observable and thus the Definition 3.2 poses no limit on his manipulations by the calling method. The state $s'$ is not directly observable but the RL method can use $\bar{s}'$ and $f = \mathcal{F}(s', D)$ (still defined in Algorithm 2).

The first element, $\bar{s}'$, can be used to draw new transitions with new calls to Env_step. With this ability, we can sample complete trajectories $(s_0, a_0, r_0, s_1, a_1, \ldots, s_H)$, something which is essential to RL methods. We can also leverage the generation of transitions to simulate common local planning procedures by calling the interface several times with $\bar{s}'$ and different actions (e.g. tree search).

Another step in converting an RL method using ML models to fit the interface is to code the manipulation of these ML models. To accomplish this, the function $\mathcal{F}(s', D)$ can: first, apply a learning algorithm applied on the internal dataset $D$ (composed of old transitions and ML models evaluation) to produce a trained ML model; second, evaluate this model on the state $s'$, giving $f$, to add information to $D$ and be manipulated by the RL method, as a value function evaluation for example.

For instance, in the case of a deep Q-learning method: $D$ has the role of a replay buffer of past experience; $\mathcal{F}$ is a learning algorithm that given the replay buffer $D$ and a state $s$, learns a Q-value function then applies it on $s$. We show in Appendix D that this construction allows for modeling the training of deep (Q-)value and policy functions.

The function $\mathcal{F}$ is thus used as a learning algorithm in our translation of existing RL methods to use this interface. The symmetry restriction on $\mathcal{F}$ posed in Definition 3.2 can be intuitively understood as a restriction on the learning algorithm. Specifically, our restriction demands that the learning algorithm treats the coordinates of the input state vectors symmetrically. This restriction is natural since, without a specific prior about the task at hand, we do not wish to process the different coordinates in particular ways. However, we note that in a practical setting with neural networks, the weights are initialized randomly, which can create asymmetries. Nevertheless, we prove in Appendix E that the distribution of trained neural networks has the demanded symmetries under a natural assumption on their architecture.

We stress that the symmetry restriction imposed on $\mathcal{F}$ does not imply that the learned ML model is invariant to permutations of its input coordinates. Instead, the restriction imposes a symmetry between the learned ML model and the dataset used for its training. For example, when learning a linear model, if we permute the first two coordinates in the training data, then the resulting model will also have the values of its first two parameters permuted.

In the special case where there is known prior information about the RL problem, the practitioner might want to use a Machine Learning algorithm that does not fit this restriction. For example, in Image Vision tasks Convolutional Neural Networks (CNNs) are commonly used and those do not respect our symmetry restriction. However, in the case of CNNs, these architectures usually produce a latent space over which an agnostic neural network architecture is applied. Our results will apply to this latent space. More generally, we believe that our results can apply under any sufficiently broad prior.

In Appendix D, as an example of our formalization, we show how classical RL methods can be translated to use the interface as constrained in Definition 3.2. We give four complete examples: a fitted Q-iteration method; a model-free policy gradient algorithm (Actor-Critic method); a local tree search model-based method; and a combination of tree-based planning and value function à la AlphaZero (Silver et al., 2017).

These examples are archetypes of the main categories of RL methods and cover Q-learning with the Bellman equation, Policy Optimization with policy gradients combined with a critic, and black-box model predictive

---

**Algorithm 3** A goal-conditioned algorithm.

---

**Parameters:** $N$: number of samples, $\alpha$: parameter of the learning algorithm (number of active features)

**Input:** $P$: the transition operator corresponding to the RL problem

$\quad D \leftarrow \{\}$

$\quad$ **for** $i \leftarrow 1, \ldots N$ **do**

$\quad\quad D \leftarrow D \cup \{(s_0, a_0, r_0, s_1, a_1, \ldots, s_H) \sim P^{\pi^U}\}$

$\quad g \leftarrow \arg\max_{s_H} r \quad$ s.t. $\quad (\ldots, r, s_H) \in D$

$\quad$ **for** $t \in \{0, \ldots, H-1\}$ **do**

$\quad\quad D_{GC}^t \leftarrow \{((s_t, s_H), a_t) \,|\, (\ldots, s_t, a_t, \ldots, s_H) \in D\}$

$\quad\quad f^t \leftarrow$ the result of the optimization program in Equation 1 with dataset $D_{GC}^t$ and parameter $\alpha$.

$\quad$ **output** $\pi(a|s_t) = f^t(a|s_t, g)$

---

control with tree-based search. The last example also covers a mix of model-based tree-based search and model-free RL. We refer to the documentation of Achiam (2018) for a taxonomy of RL methods. These examples also illustrate how several commonly used exploration methods fit our framework: $\epsilon$-greedy, on-policy, and optimism under uncertainty with UCB. Our proofs for these examples depict how other methods can be translated to fit our Definition 3.2.

## 4 Main theorem: limitation of black-box dynamics RL methods

We just characterized the class of black-box dynamics RL methods through the Definition 3.2. We prove in this section that these methods suffer from a computational efficiency limitation on a family of RL problems. In contrast, these problems are efficiently solved by a toy method. The toy method learns to map future states to actions and then applies that mapping to reach a detected rewarding state.

We define here this efficient toy method based on Hindsight Experience Replay (Andrychowicz et al., 2017).

The toy method, Algorithm 3, first samples a dataset of trajectories by drawing actions uniformly. From this dataset, it extracts a final state that gives maximal reward when entering it and poses this state as its goal. From the same dataset, a function is learned to predict the action taken from any state given the reached final state. Finally, the algorithm constructs a policy that follows the action predictor conditioned on the goal as the final state.

The learning algorithm for the action prediction minimizes the empirical rate of errors on the dataset. The space of functions in which we learn is a composition of a feature selection, a linear function, and a threshold (to output a binary prediction). The formal mathematical program is

$$\arg\min_{f_w \in F} \sum_{((s_t, s_H), a) \in D_{GC}^t} \mathbf{1}(f_w(\begin{bmatrix} s_t \\ s_H \end{bmatrix}) \neq a) \tag{1}$$
$$\text{s.t.} \quad \|w\|_0 \leq \alpha,$$

where $\begin{bmatrix} s_t \\ s_H \end{bmatrix}$ denotes the states concatenated in a vector in $\mathbb{R}^{2n}$, dataset $D_{GC}^t$ is defined in Algorithm 3, $F$ is the set of linear functions with a threshold, $F = \{x \in \mathbb{R}^{2n} \rightarrow \mathbf{1}(\langle w, x \rangle > 0) \,|\, w \in \mathbb{R}^{2n}\}$, and $w \in \mathbb{R}^{2n}$ is the parameter associated to $f_w$. The condition $\|w\|_0 \leq \alpha$ bounds the number of non-zero weights in the linear function by $\alpha \in \mathbb{N}$.

This method is elementary and limited: exploration is performed with a uniform policy; $F$ is a restricted set of functions; the method is only trying to reach a final rewarding state to maximize expected returns; and the scheme is not sound in stochastic dynamics. Thus, this method does not address the general problem of RL. Nevertheless, the method is sufficient to support the claims of this paper:

- The RL problems in Theorem 4.1 are easy to solve and not efficiently solving them is a limitation since any RL method that cannot efficiently solve these problems is beaten by Algorithm 3, a toy method that merely predicts actions to reach a detected rewarding state.

- A method that constructs a model of the inverse dynamics (such as Algorithm 3) can help to avoid the issue identified in this paper.

We have now defined everything needed to state our main result.

**Theorem 4.1.** *There exists a family of RL problems such that*

1. *For any RL problem in the family and $\delta \in (0, 1)$, with probability at least $1 - \delta$ (over the sampled trajectories), Algorithm 3 outputs an optimal policy with a number of samples and number of operations upper bounded by a polynomial in horizon $H$ and $1/\delta$.*

2. *For any algorithm satisfying Definition 3.2 and using $o(2^H)$ calls to the interface (Algorithm 2), there exists a problem in the family for which it outputs a suboptimal policy with probability at least $1/3$.*

The second affirmation bounds from below the computational complexity of black-box dynamics RL methods to solve the problems, thus for those methods the family of problems is intractable. Conversely, the first claim ensures that Algorithm 3 efficiently solves the family of problems. Thus, the Theorem proves a separation in efficiency to the disadvantage of black-box dynamics methods, implying a limitation on a large class of methods (which includes, as seen in Section 3, Q-learning, Actor-Critic methods, AlphaZero, ...). Another consequence of the Theorem is that the toy method based on Hindsight Experience Replay (Algorithm 3) cannot be formulated as a black-box dynamics method.

The complete proof is given in Appendix A. It is partially inspired by some of the proofs in Sun et al. (2019). We provide a sketch here with the main intuitions.

We explicitly construct a family of RL problems satisfying the requirements in Theorem 4.1. The problems in our family are defined by two parameters: a horizon $H \in \mathbb{N}$; a hidden binary word $b \in \{0, 1\}^{H-2}$ that represents a sequence of actions that solves the task. A representation of one element of the family for a small horizon is given in Figure 2.

The dynamics starts at a unique initial state from which two possible states are drawn randomly. The first state starts the left-hand dynamics, a unique trajectory that ends up in a rewarding state. The second state starts the right-hand dynamics, in it the number of possible trajectories grows exponentially with the horizon. At the end of these trajectories, the dynamics transitions to a final state that represents the differences between the history of actions taken and the hidden binary word $b$. If there are no differences between the binary actions taken and $b$, we arrive at the same state as the left-hand dynamics, and a reward is obtained.

The right-hand dynamics has a number of possible final states that grows exponentially in the horizon, and the unique rewarding state is hard to reach by purely random actions independent of the hidden binary word $b$. Information about $b$ is only present in the final states, and we show in the proof that the information in the final states of the right-hand dynamics is hidden under the Definition 3.2. More precisely, Definition 3.2 only allows information about these states to filter through evaluations of $\mathcal{F}(., D)$, and any $\mathcal{F}(., D)$ is provably constant on these states due to the symmetry restriction. Methods satisfying Definition 3.2 will thus not do better than enumerating all the possible behaviors since they do not have access to relevant information to orient the search in the right-hand dynamics.

We illustrate the reasoning with the example of a Q-learning algorithm implemented with the interface applied to the problem in Figure 2. When sampling some final state $s_H$, the only information filtering through the interface to the method is the obtained reward, the encoding $\bar{s}_H$, and $Q_{s_H} = \mathcal{F}(s_H, D)$. The encoding $\bar{s}_H$ is simply the number of states encoded up to now and reveals no information about $s_H$ and the problem. The output $Q_{s_H}$ represents the Q-value estimates at that state built from the dataset $D$ composed of tuples $(s, a, r, Q_{s'})$, where these tuples are based on previously sampled transitions $(s, a, r, s')$ and an old

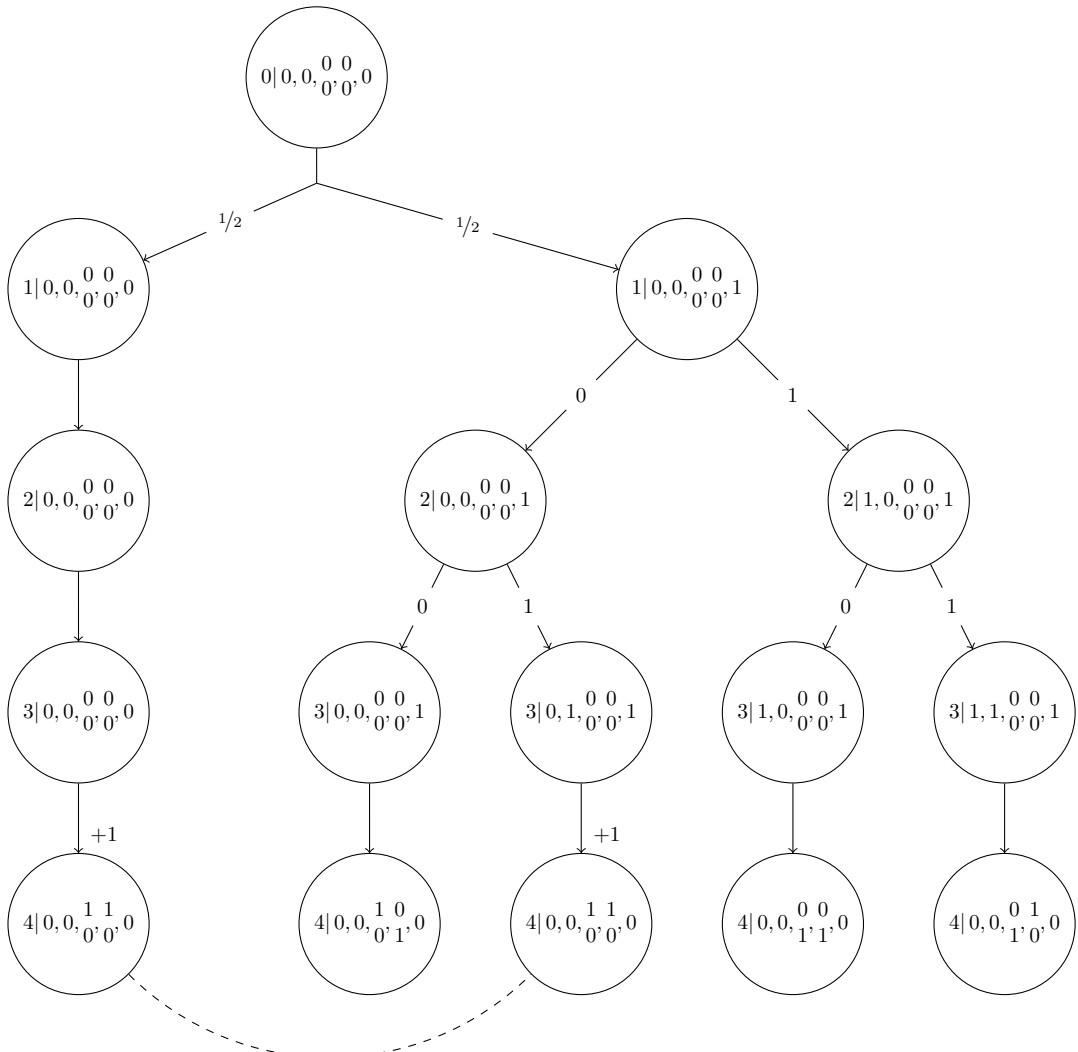

Figure 2: Example of the family of RL problems for horizon $H = 4$ and hidden binary word $b = 01$. On the left-hand dynamics, the agent can easily discover the unique rewarding state. In comparison, a reward is hard to get in the right-hand dynamics using random exploration but by exploiting the dynamics and a discovered relevant goal, an algorithm can uncover an optimal path (such as toy method Algorithm 3).

estimation $Q_{s'}$ of the Q-values at $s'$. The dataset can only contain non-final states by construction of the interface. Also, by construction of the problem, the (non-final) states in the dataset and final states have different coordinates with non-zero elements. Given these elements, an analysis with the symmetry condition on $\mathcal{F}$ in Definition 3.2 reveals that $Q_{s_H} = \mathcal{F}(s_H, D)$ is constant with respect to $s_H$. Thus, in this case, only the obtained reward exposes information about the problem for a black-box dynamics method.

In this analysis, the symmetry condition asked on $\mathcal{F}$ is crucial. It allows us to prove that the outputs of $\mathcal{F}$ are equal for different states in our problems. These states are thus indistinguishable, and imply that the algorithm fails to leverage the crucial information in them.

For Algorithm 3, the left-hand dynamics allows the RL method to discover the rewarding state easily and set it as its goal. The right-hand dynamics is sufficiently simple for the learning procedure to reliably predict the necessary sequence of actions to reach this goal. With high probability, the returned policy will thus reach the rewarding state in the right-hand dynamics.

**Generalization** Our result defines an example family of problems but does not provide general conditions on problems upon which the limitation on black-box dynamics methods appears. Nevertheless, from this family, we can extract some key elements to our proof of the limitation:

- the dynamics have several actions to take and each is critical to the success;

- each action has a positive or negative effect depending on the initially unknown dynamics;

- the dynamics aggregate the effects of all the actions into one binary reward, thus hiding the effects of each action to a method relying on reward feedback.

At the same time, these problems are tractable for our toy methods because:

- a rewarding state reachable with the above actions can be inferred from random exploration;

- the effect of each action is revealed in the state-to-state transitions.

These core characteristics informally define sufficient conditions to show a limitation. An improvement over our result could be to formally define sufficient conditions as general as possible on problems for the occurrence of a limitation. This would require several generalizations over Theorem 4.1. More precisely, it would be necessary to generalize the bound on the performance of black-box dynamics methods to a broader range of problems. Additionally, a more universal algorithm should be proposed than our toy method (Algorithm 3), supported by an equally universal guarantee of its performance. These generalizations and their scope will determine the breadth of any obtained limitation.

**Domination** A corollary of Theorem 4.1 is that for any black-box dynamics RL method, another method exists that always performs at least as well and sometimes better on RL problems. This result can be obtained by constructing a new method that executes in parallel both the black-box dynamics method and Algorithm 3. Appendix A contains a formal statement and associated derivation of this fact.

## 5 Numerical experiments

We illustrate and confirm numerically how practical deep RL methods perform on the family of RL problems constructed in the proof of Theorem 4.1. We also test the goal-conditioned method from the last section. Moreover, we introduce another method that performs well on our set of problems. This method leverages a non-black-box model of the dynamics for planning.

We test model-free methods such as fitted Q-iteration (Riedmiller, 2005). We implement and run a classical Actor-Critic method, Proximal Policy Optimization (PPO) (Schulman et al., 2017). Moreover, we test AlphaZero, a model-based method leveraging learning and local planning (Silver et al., 2017). We refer to Appendix G for details on the implementations.

We test Algorithm 3 but with neural networks trained by gradient descent as a learning procedure instead of the function space defined in Equation 1.

In addition, we propose and test a second algorithm that empirically performs well on the family of RL problems. A more detailed description of this algorithm is available in Appendix F.

This algorithm iteratively alternates between two computations. One, learning a model of the dynamics from sampled trajectories. Two, sample trajectories by using a planning algorithm to determine the actions. The planning algorithm converts the current state and the learned model of the dynamics into a mathematical program that maximizes the sum of future rewards. Then, an off-the-shelf solver solves the mathematical program and outputs a plan with the action to take. The algorithm iterates this procedure until a performance threshold is obtained, then the training stops and the method outputs the planning procedure as its policy.

We propose this algorithm to support two claims.

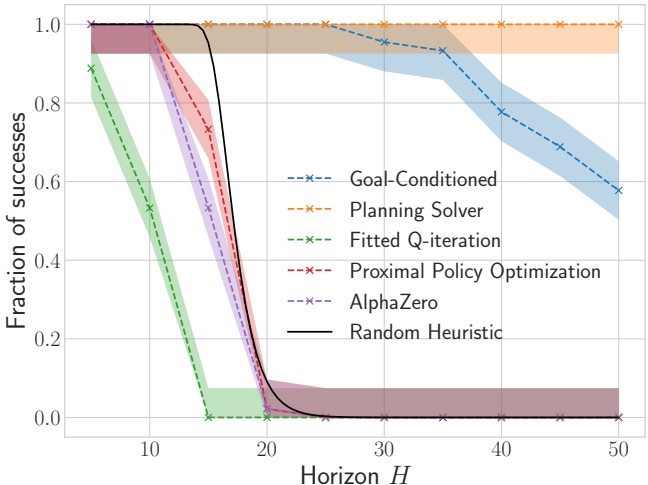

Figure 3: Fraction of successes as a function of the horizon for each method on the RL problem family illustrated in Figure 2. We performed 45 runs for each point on the Figure. An upper bound on the standard deviation is displayed. The *Goal-Conditioned* denotes Algorithm 3 with neural networks and *Planning Solver* is the planning method presented in this section. The *Random Heuristic* denotes the analytically computed performance of a method that samples random actions until reaching a rewarding state on the right-hand dynamics of the problems, then copies that behavior. This method assumes access to 50.000 sampled trajectories.

Firstly, we illustrate in a second way that our RL problems are tractable. This method relies on the fact that the dynamics of our MDPs are easy to learn and an off-the-shelf solver can plan efficiently in these dynamics. Those are general properties that an efficient general method should be able to take advantage of.

Secondly, this algorithm illustrates another idea in the literature than the goal-conditioned Algorithm 3. Algorithm 3 learns a link from future states to present actions directly, while here we propose an algorithm that learns a model of the forward dynamics. This algorithm is also interesting to compare with other model-based RL methods on which the limitation applies (e.g. based on tree search). This new planning algorithm distinctively leverages an explicit model of the dynamics by decomposing this model into structured constraints, instead of only using the model as a black-box generator of samples.

We apply these methods to problems of the family constructed in the proof of Theorem 4.1 with an increasing horizon. For the fitted Q-iteration and PPO methods, we sample 50 times 1000 trajectories during one training run. The AlphaZero method accesses 3 times 1000 trajectories, where for each state in these trajectories, the planning procedure leverages 50 simulation of transitions. Thus, the method leverages an equivalent total of 150 000 sampled trajectories. We sample a dataset of 1000 trajectories for the goal-conditioned method. The new planning algorithm takes 1000 randomly sampled trajectories for its initialization, and then at most 1000 additional trajectories are sampled in the alternations between learning and planning for any run.

To measure the success of a method, we check if its returned policy reaches the rewarding state in the right-hand dynamics with 1000 sampled trajectories, this allows the goal-conditioned algorithm to do minimal exploration.

The results are presented in Figure 3. The model-free methods quickly fail to solve the task when the horizon is increased, their performance is bounded by a simple heuristic method that randomly explores the environment and then copies the best-performing behavior. This is in line with our theoretical results.

Despite being allowed fewer samples and not more computing time, the goal-conditioned and planning methods successfully solve the tasks for much longer horizons.

## 6 RL methods avoiding the limitation

There exist several types of methods that do not fall in the class described by Definition 3.2, and are thus potentially free of the flaw highlighted in this paper. We identify the following algorithmic approaches with that property in the RL literature.

**Backpropagation through a learned model.** The algorithms learn a smooth model of the dynamics then use backpropagation through the learned model (Nguyen & Widrow, 1990; Jordan & Rumelhart, 2013; Deisenroth & Rasmussen, 2011; Grondman, 2015; Heess et al., 2015). These methods elude our definition because the learned model is not treated as a black-box by the algorithm. We note that these methods are mostly used in environments where the dynamics are approximately smooth, such as low-level control in robotics.

**Classical/symbolic planning.** The algorithms construct a symbolic or semi-parametric model of the dynamics and then apply a classical planning algorithm (Russell & Norvig, 2010; Konidaris et al., 2018; Kurutach et al., 2018; Asai et al., 2022). This example is related to the previous one, it evades the proved limitation when the symbolic planner does not treat the model only as a black-box generator of transitions but leverages insights in it. The method presented in the numerical experiment section is an implementation of this idea since it decomposes its learned model of the dynamics into structured constraints for planning.

**Universal value functions.** Universal value functions (Sutton et al., 2011; Schaul et al., 2015; Andrychowicz et al., 2017) can be learned to decide which states can be efficiently reached from which states. These algorithms bypass our definition because the value functions take as input the states in the future of the trajectory, and the quantities they learn cannot be replaced trivially by a generator of transitions.

**Inverse model and goal-conditioned.** A wide variety of algorithms learn a link from the future to the past of a trajectory. An example is learning an inverse dynamics model (Mordatch & Hamrick, 2020). These algorithms avoid the limitation for the same reason as the universal value functions: the future of the trajectory is directly used to learn the relevant functions, and there is no trivial efficient way to replace the computed quantities with a generator of transitions.

Inverse dynamics models or goal-conditioned methods learn to predict the action to take given the current state and a state to reach in the future (Ghosh et al., 2019). There exist variants of this idea, Janner et al. (2022) propose to learn a mapping from a current and a future state to a full sequence of intermediary states.

These different functions can be efficiently learned with Hindsight Experience Replay, where what is reached in a trajectory is relabelled as a goal in hindsight for training (Kaelbling, 1993; Andrychowicz et al., 2017).

We note however that not all these methods are sound in the presence of uncertainty, as described (and alleviated) in Paster et al. (2022); Eysenbach et al. (2022); Yang et al. (2022); Villaflor et al. (2022).

**Exploration methods** We refer to Ladosz et al. (2022) for a survey on exploration in RL methods and follow their taxonomy. Our class of black-box dynamics methods covers some commonly used exploration techniques such as $\epsilon$-greedy and optimism under uncertainty with UCB, as shown in Appendix D. Other exploration techniques do not necessarily fit our class. Methods based on maximizing intrinsic rewards, state or behavior diversity, do not solely focus on the task-reward and are thus not covered by Definition 3.2. Also, methods based on exploratory goals might use universal value functions and Hindsight Experience Replay approaches which are also out of our class, as discussed above.

Not all of these algorithms necessarily solve efficiently the family of RL problems we defined. Our theoretical and numerical results suggest that some ideas present in them could help solve problems otherwise intractable for a large class of classical RL methods.

# 7   Conclusion

We introduced the class of black-box dynamics RL methods with Definition 3.2 and its linked interface. This class encompasses model-free methods such as Q-learning and Policy Optimization, as well as several model-based methods such as AlphaZero. For this broad range of methods, we proved a computational efficiency limitation on a family of problems in Theorem 4.1. However, we described two toy methods that can efficiently solve these problems.

The problems used in the proof of Theorem 4.1 feature two different dynamics randomly chosen at the start of the trajectory. The first dynamics is simple and allows an algorithm to easily discover a rewarding state. In the second dynamics, finding a behavior that reaches a rewarding state is hard. However, these two dynamics have a common unique rewarding state. An efficient method is able to discover the rewarding state in the first dynamics and then plan to reach that state with learned knowledge of the second dynamics.

To summarize our findings intuitively, a large class of RL methods will unnecessarily struggle in environments where following rewards alone is not informative enough. Our results reveal an incapacity of black-box dynamics methods to leverage essential information from some RL problem dynamics. In Section 6, we list several ideas in the literature that could help to construct general-purpose RL methods without the identified flaw.

## Acknowledgements

RJ is a FNRS honorary Research Associate. This project has received funding from the European Research Council (ERC) under the European Union's Horizon 2020 research and innovation program under grant agreement No 864017 - L2C, from the Horizon Europe program under grant agreement No101177842 - Unimaas, and from the ARC (French Community of Belgium)- project name: SIDDARTA.

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

## A    Main Theorem proof

In this section, we provide the main Theorem proof, then compare it with the proof in Sun et al. (2019). Finally, we state and prove a domination result inferred from our main Theorem.

**Theorem 4.1.** *There exists a family of RL problems such that*

1. *For any RL problem in the family and $\delta \in (0, 1)$, with probability at least $1 - \delta$ (over the sampled trajectories), Algorithm 3 outputs an optimal policy with a number of samples and number of operations upper bounded by a polynomial in horizon $H$ and $1/\delta$.*

2. *For any algorithm satisfying Definition 3.2 and using $o(2^H)$ calls to the interface (Algorithm 2), there exists a problem in the family for which it outputs a suboptimal policy with probability at least $1/3$.*

*Proof.* We define the family of RL problems. The problems in the family are parametrized by a horizon $H$ and by a hidden binary word $b$ of length $H - 2$. Each state is defined by $3(H - 2) + 1$ real variables. The action space is binary. An example in this family for $H = 4$ and $b = 01$ is represented in Figure 2.

We pose some notations to describe the states. Each state is decomposed into three parts $a$, $b$, and $c$, the part $b$ is further decomposed into parts $u$ and $d$. The time step to which the state belongs is kept implicit in our description. For $i$ between 1 and $H - 2$:

- $s^{a,i}$ the $i$th variable in the first part of the state vector;

- $s^{b,u/d,i}$ if $u$, the up $i$th variable in the second part of the state vector, if $d$, the down $i$th variable in the second part of the state vector;

- $s^c$ the variable of the third part of the state vector.

Example of the notation for state vector $s$:

$$\left[ s^{a,1}, s^{a,2}, \ldots, s^{a,i}, \ldots, s^{a,H-2}, \begin{matrix} s^{b,u,1} & s^{b,u,2} \\ s^{b,d,1}, & s^{b,d,2} \end{matrix}, \ldots, \begin{matrix} s^{b,u,i} \\ s^{b,d,i} \end{matrix}, \ldots, \begin{matrix} s^{b,u,H-2} \\ s^{b,d,H-2} \end{matrix}, s^c \right].$$

Now we describe the dynamics. At step 0, whatever action is taken, with probability $1/2$, we reach either a state full of zeros, or a state full of zeros except for $s^c$ which equals one.

If we have $s^c = 0$, then for the steps $t = 1$ to $t = H - 1$, the state vector stays zero. At the last step it gives a reward of 1 and the state with $s^a = 0^{H-2}$, $s^{b,u} = 1^{H-2}$, $s^{b,d} = 0^{H-2}$ and $s^c = 0$.

If $s^c = 1$, then from time step $t = 1$ to $t = H - 2$, the transition from $t$ to $t + 1$ encodes the binary action taken into $s^{a,t}_{t+1}$ with zero reward (the other variables keep their values from the previous time step). At the last step, the dynamics fixes $s^c_H = 0$, all the variables that encode a past action to zero $s^a_H = 0^{H-2}$, and $s^b_H$ to express if each action taken was correct $\begin{pmatrix} 1 \\ 0 \end{pmatrix}$ or not $\begin{pmatrix} 0 \\ 1 \end{pmatrix}$ with respect to a fixed optimal trajectory given by $b$ the binary word parametrizing the problem. In other words, $s^{b,u,i}_H = \mathbf{1}(s^{a,i}_t = b_i)$ and $s^{b,d,i}_H = \mathbf{1}(s^{a,i}_t \neq b_i)$. At that step, if $s^{b,u}_H = 1^{H-2}$, then a reward of 1 is given. Such that, a reward is obtained only if all the actions taken follow the hidden binary word $b$.

We prove the first claim of the Theorem.

Take Algorithm 3 and fix any $\delta \in (0, 1)$.

There exists a lower-bound polynomial in $\log 1/\delta$ on the number of sampled trajectories, $K$, such that for any horizon, $H$, with probability at least $1 - \delta/2$ the rewarding state is discovered with the left-hand dynamics $(s^c = 0)$.

Moreover, we demonstrate in Lemma A.6, that Algorithm 3 obtains, for $\alpha = 1$ and any $K$ larger than some polynomial in $H$ and $1/\delta$, with probability at least $1 - \delta/2$ over the sampled dataset, a goal-conditioned function which predicts correctly the action to take on the right-hand dynamics (when $s^c = 1$).

By choosing $K$ sufficiently large and applying the union bound, the probability that one of these events fails is bounded by $\delta$. With high probability, the policy $\pi(a|s_t) = f^t(a|s_t, g)$ will reach the rewarding state (and thus act optimally) when $s_t^c = 1$. When $s_t^c = 0$, the policy does not influence the expected return. The policy is thus optimal.

Moreover, this method can be efficiently implemented from a computational complexity point of view.

Now we prove the second claim. We will prove that the part of the problem where $s^c$ equals one is equivalent to an armed-bandit problem family (Definition A.7) with $2^{H-2}$ arms where only one arm gives a reward. Following Proposition A.8, any algorithm using $o(2^H)$ calls will fail to identify the best arm with probability at least $1/3$ on some problem in the armed-bandit problems family. This entails the Theorem.

In the rest of the proof, we assume that we are on the branch where $s^c = 1$, the branch $s^c = 0$ is constant across the family for a fixed $H$ and thus provides no information on $b$.

We prove that all the states at the last steps which have zero-reward are indistinguishable. Due to Definition 3.2, information on these states can only be obtained through evaluations of a function $\mathcal{F}$ with one of these states as the first argument (since these states are final, they cannot be added as direct input in the dataset $D$).

We prove that these evaluations are identical for all those states. These states only contain non-zero elements in coordinates which are always zero in the states present in the dataset. Moreover, they have exactly the same number of ones in those coordinates. Thus, $\mathcal{F}$, which is symmetric on permutations of the coordinates, cannot distinguish them. Formally, let be two final states on the right-hand side $w, z$, there exists a permutation of the coordinates $p$ such that $w = p(z)$ and $s = p(s)$ for all states $s$ present in the dataset $D$. This implies, for any $\mathcal{F}$, by the symmetry constraint on $\mathcal{F}$,

$$\mathcal{F}(z, ((s_0, a_0, r_0, f_0), \ldots,)) = \mathcal{F}(p(z), ((p(s_0), a_0, r_0, f_0), \ldots))$$
$$= \mathcal{F}(w, ((s_0, a_0, r_0, f_0), \ldots)).$$

Thus, non-rewarding final states on the right-hand dynamics are indistinguishable for RL methods satisfying Definition 3.2. The family of right-hand dynamics is as hard as the family of armed-bandit problems for those methods.

$\square$

Similarly to Sun et al. (2019), we prove that for the studied class of methods, the problem family is indistinguishable from a hard family of black-box problems, implying the limitation. We adapt the proof to our class of black-box dynamics methods, by changing how information is hidden to prove the indistinguishability. Instead of hiding some states behind their evaluations with optimal value functions as in Sun et al. (2019), we hide them behind function evaluations corresponding to learned ML model outputs. We leverage the symmetries these models satisfy with respect to their constrained dataset to prove that some states are treated equivalently. Our interface also outputs pointers to states $\bar{s}$ but those do not disclose any information on the states.

Another point of comparison with the proof of Sun et al. (2019) is the addition of a trajectory allowing a method to discover a rewarding state. This is critical to get a method that can efficiently solve the presented family without an a priori goal-state. In parallel to this addition, the efficient method and the guarantee of its performance are also new to this proof. We refer to the introduction and Appendix C for a summary of what our results gain from these differences.

The following definitions and affirmations are classical results from VC dimension theory and can be found in the reference Shalev-Shwartz & Ben-David (2014).

**Definition A.1.** Hypothesis class.
A set $\mathcal{H}$ is a hypothesis class if $\mathcal{H}$ is a set composed of functions from some domain $\mathcal{X}$ to $\{0, 1\}$.

**Definition A.2.** Shattering.
A finite set $S \subseteq \mathcal{X}$ is shattered by a hypothesis class $\mathcal{H}$ if for any function $f$ from $S$ to $\{0, 1\}$, there exists $h \in \mathcal{H}$ such that for all $x \in S$ we have $f(x) = h(x)$.

**Definition A.3.** VC-dimension.
Given a set $\mathcal{X}$, the VC-dimension of a hypothesis class defined on domain $\mathcal{X}$ is the maximal integer $d$ such that there exists a subset of $\mathcal{X}$ of size $d$ which is shattered by $\mathcal{H}$.

**Proposition A.4.** *VC-dimension of the union.*
*Given $r$ hypothesis classes of VC-dimension at most $d$ sharing the same domain, the VC-dimension of the union of these classes is at most $4d \log(2d) + 2 \log(r)$.*

**Proposition A.5.** *Agnostic PAC-learnability from bounded VC-dimension.*
*There exists a constant $C_1 \in \mathbb{R}$ such that for any $\delta, \epsilon \in (0, 1)$, given a hypothesis class $\mathcal{H}$ of VC-dimension $d$, and a dataset $S \subset \mathcal{X} \times \{0, 1\}$ of size at least $C_1 \frac{d + \log(1/\delta)}{\epsilon^2}$ composed of identically and independently drawn samples $(x, y)$ according to a probability measure $P$ on $\mathcal{X} \times \{0, 1\}$, we have, with probability at least $1 - \delta$ (over the sampled dataset),*

$$L_P(\arg\min_{h \in \mathcal{H}} L_S(h)) \leq \min_{h' \in \mathcal{H}} L_P(h') + \epsilon,$$

*where $L_P(h)$ is the error rate of classification according to measure $P$, $\mathbb{E}_{(x,y) \sim P}[\mathbf{1}(h(x) \neq y)]$, and $L_S(h)$ is the error rate on the dataset, $\frac{1}{|S|} \sum_{(x,y) \in S} \mathbf{1}(h(x) \neq y)$.*

**Lemma A.6.** *For Algorithm 3 with parameter $\alpha = 1$ and any $\delta \in (0, 1)$ on the family presented in the proof of Theorem 4.1, there exists $m$ a polynomial in $H$ and $1/\delta$, such that with a dataset of trajectories larger than $m(H, 1/\delta)$ obtained by $\pi^U$, with probability at least $1 - \delta$ over the sampled dataset, for all time steps $1 \leq t \leq H - 2$ and with $s_t^c = 1$, the learned function $f^t(a | s_t, g)$ perfectly predicts the action to reach the goal $g$, if the state $g$ is reachable from $s_t$.*

*Proof.* The proof can be decomposed in 4 main affirmations:

1. The classes of learning hypotheses have a VC-dimension upper-bounded by a polynomial in $H$.

2. For all the problems in the family, there exists a hypothesis with a low rate of errors.

3. A sufficiently low error rate implies a perfect accuracy on the predicted action to reach $g$ when $s_t^c = 1$.

4. Set the last three points together to entail the Lemma.

We pose $s_t$ as the current state, $s_H$ as the state at the end of the trajectory after observing $s_t$. Thus, the learned function takes as input the vector $\begin{bmatrix} s_t \\ s_H \end{bmatrix}$ (concatenation of the real parts of the states without the time steps).

**1.** For $\alpha = 1$, at any time step, the class of hypotheses is the union of $O(H)$ sets of linear functions over 1 real-valued variable composed with a threshold. Those linear functions with a threshold have thus VC-dimension 2. Using Proposition A.4, we infer that, for any time step, the VC-dimension of our hypothesis class is in $O(\log H)$.

**2.** Take the hypothesis that selects the variable $s_H^{b,u,t}$ and uses the linear identity function composed with a threshold $\mathbf{1}(s_H^{b,u,t} > 0)$.

Both sides of the environment ($s^c = 0$ or 1) have $1/2$ probability. Since all the samples on the right-hand dynamics are correctly predicted by this hypothesis and the samples on the left-hand side have $1/2$ probability to have either action, this hypothesis has an average error rate of $1/4$ on the distribution induced by $\pi^U$.

**3.** For all the possible feature selections, there is a finite set of possible inputs, for example, $\{0, 1\}$ for feature $s_H^{b,u,i}$, or $\{0\}$ for feature $s_t^{b,d,i}$. Conditioned on $s_t^c = 1$, these inputs all have a constant probability of being sampled by $\pi^U$, thus the probability is independent of $H$ and $b$ the family parameters. We note the minimum on these probabilities $p_{\min}^{\text{input}}$, which is thus also independent of the family parameters $H$ and $b$.

Similarly, we note $p_{\min}^{\text{output}}$ the minimal non-zero probability under $\pi^U$ of any of the two actions conditioned on the selected feature from a state in the right-hand dynamics.

Suppose that the error rate under the distribution induced by $\pi^U$ of hypothesis $h$ for the goal-conditioned prediction problem is $1/4 + \epsilon$. If $\epsilon < 1/2 \cdot p_{\min}^{\text{input}} \cdot p_{\min}^{\text{output}}$ then the hypothesis does not make any errors in the right-hand dynamics. By contradiction, suppose that it does make a mistake on a pair of input-output, then take the value of the input of the variable selected by the hypothesis. There is at least $1/2 \cdot p_{\min}^{\text{input}}$ samples with that input and a $p_{\min}^{\text{output}}$ proportion of them with the same output. The hypothesis will thus make a mistake on a set of inputs of probability at least $1/2 \cdot p_{\min}^{\text{input}} \cdot p_{\min}^{\text{output}}$ under $\pi^U$ on the right-hand dynamics. To which we have to add a $1/2$ error rate on the trajectories on the left-hand dynamics that have probability $1/2$.

**4.** To generalize the result to all the time steps at the same time, we use a union bound over the probability of failure of each time step.

Using Proposition A.5 and the affirmations just proven, with any constant $\epsilon < p_{\min}^{\text{input}} \cdot p_{\min}^{\text{output}}/2$ and $\delta = \delta/H$, there exists $m(H, \delta) \in O(\log \frac{H}{\delta})$ such that the algorithm will output a hypothesis which makes no mistake on the right-hand dynamics. $\qquad\square$

**Definition A.7.** A *deterministic armed-bandit problem* is composed of a finite set of arms $\mathcal{A}$ and a function $f : \mathcal{A} \to [0, 1]$ which associates an arm to a deterministic reward.

**Proposition A.8.** *Let be a family of deterministic armed-bandit problems defined by $A \in \mathbb{N}$ and $1 \leq i^* \leq A$. The deterministic armed-bandit problem defined by $A$ and $i^*$ has $A$ arms, the arm $i^*$ gives a reward of $1$, and the others give a reward of $0$.*

*For any randomized algorithm that tries to identify the arm with the maximal associated reward with $o(A)$ calls to the reward associating function, there exists an $A$ large enough and a $i^*$ such that the algorithm fails with probability at least $1/3$ on the problem defined by $A$ and $i^*$.*

*Proof.* By contradiction, let's suppose that there exists a randomized algorithm Alg that solves any problem in the family with probability at least $1 - \delta$ for some $\delta \in (0, 1)$ using $K \in o(A)$ samples. Let us note $r \sim R$ the random variable following some probability distribution $R$ upon which the algorithm depends, $p$ the problem in input that it solves, and $\text{Alg}(p, r)$ the output of the algorithm with $p$ and $r$ in input. For any $A$, let $U(A)$ be the uniform probability distribution upon $\{1, \dots, A\}$ and $p_{i^*}^A$ the bandit problem defined by $A$ and $i^*$. We have for any $A$

$$\mathbb{E}_{i^* \sim U(A)}\mathbb{E}_{r \sim R}[\mathbf{1}(\text{Alg}(p_{i^*}^A, r) = i^*)] \geq 1 - \delta. \tag{2}$$

Which implies

$$\mathbb{E}_{r \sim R}\mathbb{E}_{i^* \sim U(A)}[\mathbf{1}(\text{Alg}(p_{i^*}^A, r) = i^*)] \geq 1 - \delta. \tag{3}$$

There must exist some $r$ which performs at least as well as the mean. Thus there exists a deterministic algorithm $\text{Alg}_d$ such that $\mathbb{E}_{i^* \sim U(A)}[\text{Alg}_d(p_{i^*}^A)] \geq 1 - \delta$.

We suppose w.l.o.g. that this algorithm uses its $K \in o(A)$ tries in the set of the $K$ first arms. Then with probability $1 - \frac{K}{A}$ it only receives $0$ rewards and must guess the best arm with no information for the $A - K$ left untested arms. Thus the deterministic algorithm has a probability of failure lower bounded by $1 - \frac{K}{A} - \frac{1}{A-K}$.

Since $K \in o(A)$, there exists $A$ large enough such that this quantity is larger than $\delta$. Thus we have a contradiction. $\qquad\square$

### A.1 Domination result

Here we derive from Theorem 4.1 a result on the domination of black-box dynamics RL methods.

We define $u(\pi, m) = \mathbb{E}_{(s_0, a_0, r_0, \ldots, s_H) \sim P^\pi} [\sum_{t=0}^{H-1} r_t]$ the expected cumulative reward of some policy $\pi$ on RL problem $m$ with associated operator $P$. Additionally, we define $U(A, m)$ the average expected cumulative reward of (randomized) RL method $A$ on RL problem $m$. More formally, with $\pi$ the (random) output of $A$ on problem $m$, $U(A, m) = \mathbb{E}_\pi [u(\pi, m)]$.

**Corollary A.9.** *For any RL method $A$ taking at most polynomial time in the horizon of the problem and satisfying Definition 3.2 there exists an RL method $B$ that takes at most polynomial time in the horizon and $1/\epsilon$ with $\epsilon \in (0, 1)$ such that*

1. *For any RL problem $m$, $U(A, m) \leq U(B, m) + \epsilon$;*

2. *There exists an RL problem $m$ such that $U(A, m) \leq U(B, m) - 1/12$.*

*Conversely, there exists some $\epsilon \in (0, 1)$ and an RL method $B$ taking at most polynomial time in the horizon of the problem such that there does not exist a method $A$ satisfying Definition 3.2 and that for any RL problem $m$ satisfies $U(B, m) \leq U(A, m) + \epsilon$.*

*Proof.* For any RL method $A$ satisfying the conditions of the statement, construct the RL method $B$ performing the following computations for an RL problem of horizon $H$:

1. Find policy $\pi_{GC}$ by executing Algorithm 3 with parameters $\alpha = 1$ and $N$ sufficiently large to have a maximal failure probability of $\delta_0 = 1/24$ for problems in the family in the proof of Theorem 4.1 with horizon $H$.

2. Evaluate the performance of $\pi_{GC}$ with an accuracy of $\min\{\epsilon/4, 1/20\}$ with a success probability of at least $1 - \delta/2$ with $\delta = \min\{\epsilon/2H, 1/24\}$.

3. Execute method $A$ on the RL problem, take output policy $\pi_A$.

4. Evaluate the performance of $\pi_A$ with an accuracy of $\min\{\epsilon/4, 1/20\}$ with a success probability of at least $1 - \delta/2$ with $\delta = \min\{\epsilon/2H, 1/24\}$.

5. Return the policy with the largest estimated performance, call it $\pi_B$.

Step 1 takes polynomial time in the relevant quantities, we refer to the proof of Theorem 4.1. Step 3 is also efficient by the condition on method $A$. Steps 2 and 4 can be performed in polynomial time in the relevant quantities using Hoeffding's inequality. Thus the whole method is efficient (polynomial in the horizon and $1/\epsilon$).

For the first claim, consider $m$ an input problem. By construction the policy returned by method $B$ has expected cumulative rewards at least of $\max\{u(\pi_{GC}, m), u(\pi_A, m)\} - \epsilon/2$ with probability $1 - \delta$. Thus we have the following lower-bound on the performance

$$
\begin{aligned}
u(\pi_B, m) &\geq (1 - \delta)(\max\{u(\pi_{GC}, m), u(\pi_A, m)\} - \epsilon/2) + \delta \cdot 0 \\
&\geq (1 - \delta)(u(\pi_A, m) - \epsilon/2) \\
&\geq u(\pi_A, m) - \delta u(\pi_A, m) + \delta\epsilon/2 - \epsilon/2 \\
&\geq u(\pi_A, m) - \delta H - \epsilon/2 \\
&\geq u(\pi_A, m) - \epsilon.
\end{aligned}
\tag{4}
$$

We used the fact that the rewards are constrained in the interval $[0, 1]$ and thus the expected cumulative rewards are upper-bounded by $H$ and lower-bounded by 0.

Since the probability distribution of $\pi_A$ produced in method B is the same as the output of method A, we have

$$U(B, m) \geq E_{\pi_B}[u(\pi_B, m)] \geq E_{\pi_A}[u(\pi_A, m)] - \epsilon = U(A, m) - \epsilon. \tag{5}$$

This concludes the proof of the first claim.

For the second claim, method $A$ satisfies Definition 3.2 and has computational complexity in $o(2^H)$. Thus method A satisfies the conditions of the second claim in Theorem 4.1. From the proof of Theorem 4.1, we can extract the fact that $U(A, m)$ is at most $5/6$ for some RL problem $m$ in the family we defined[1].

While, by construction $\pi_{GC}$ outputs an optimal policy with probability $1 - \delta_0$. Moreover, if $\pi_{GC}$ is optimal, it is chosen as output $\pi_B$ with probability at least $1 - \delta$ since the asked accuracy $1/20$ is below half the difference in expected returns for $\pi_{GC}$ and $\pi_A$ (lower-bounded by $1/12$).

Thus the expected returns of $\pi_B$ satisfies, taking into account that for the problems in the family 1 is the optimal expected return achievable,

$$\begin{aligned} u(\pi_B, m) &\geq (1 - \delta - \delta_0) \cdot 1 + (\delta + \delta_0) \cdot 0 \\ &= 1 - 1/12. \end{aligned} \tag{6}$$

Thus,

$$U(B, m) - 1/12 \geq E_{\pi_B}[u(\pi_B, m)] - 1/12 \geq 1 - 1/12 - 1/12 \geq 1 - 1/6 \geq U(A, m). \tag{7}$$

The converse and last statement is a direct consequence of Theorem 4.1 proof by taking Algorithm 3 as $B$. □

# B  Parallel result with a different assumption

This section presents another formalism than the main text, this formalism is closer to the paper of Sun et al. (2019). We derive a parallel version of our theoretical result in this new formalism. Then, we discuss the advantages and inconveniences of this formalism in contrast to the one used in our main text.

This section aims to clarify the characteristics of different formalization choices. A second goal is to show that the ideas presented in this paper hold in the two formalisms; thus, their consequences are broader than what the main text formally presents.

## B.1  The formalism and result of Sun et al. (2019)

We provide here briefly the result of Sun et al. (2019).

The following definition of a model-free method is given by Sun et al. (2019).

**Definition B.1.** Model-free algorithm, Sun et al. (2019). Given a (finite) function class $\mathcal{G} : (\mathcal{S} \times \mathcal{A}) \to \mathbb{R}$, define the $\mathcal{G}$-profile $\phi_{\mathcal{G}} : \mathcal{S} \to \mathbb{R}^{|\mathcal{G}| \times |\mathcal{A}|}$ by $\phi_{\mathcal{G}}(x) \overset{\text{def}}{=} [g(x, a)]_{g \in \mathcal{G}, a \in \mathcal{A}}$. An RL method is model-free using $\mathcal{G}$ if it accesses $x$ exclusively through $\phi_{\mathcal{G}}$ for all $x \in \mathcal{S}$ during its entire execution.

The definition of a model-free method depends on a set of functions $\mathcal{G}$. For their theorem, they assume $\mathcal{G} = \text{OP}(\mathcal{M})$, where OP stands for optimal planning, it is defined as the set of optimal Q-functions and policy functions for a given family of RL problems $\mathcal{M}$. In other words, the set $\text{OP}(\mathcal{M})$ contains any policy and Q-function that is optimal for some problem in $\mathcal{M}$.

Depending on the family $\mathcal{M}$, this assumption implies a barrier to the information that can go from the encountered states to the model-free method. This loss of information allows them to prove their theorem on the gap in statistical complexity between a model-based method and model-free RL methods.

The work of Sun et al. (2019) proves that there exists a family of RL problems $\mathcal{M}$ such that:

---

[1]The method $A$ has less than a $2/3$ probability to produce an optimal sequence of actions on the right-hand dynamics. Taking into account the two sides of the dynamics, the average expected cumulative rewards is thus at most $1/2 \cdot 1 + 1/2 \cdot 2/3 = 5/6$

---

**Algorithm 4** Env : an interface linked to a RL problem defined by the transition operator $P$ and a $\mathcal{G}$-profile $\phi_{\mathcal{G}}$ (Definition B.1). The interface is a set of functions with an internal state. The interface has access to an initialized encoder/decoder of pointers and states as defined in Algorithm 1.

---

    **function** Env_init($\mathcal{F}$)
        $s_0 \sim P_0$
        $\bar{s}_0 \leftarrow$ encode_state($s_0$)
        **output** $\bar{s}_0$, $\phi_{\mathcal{G}}(s_0)$
    **function** Env_step($\bar{s}$, $a$)
        $s \leftarrow$ decode_pointer($\bar{s}$)
        **if** $s$ is not terminal **then**
            $r, s' \sim P_{\mathrm{dyn}}(r, s' \,|\, s, a)$
            $\bar{s}' \leftarrow$ encode_state($s'$)
            **output** $r$, $\bar{s}'$, $\phi_{\mathcal{G}}(s')$
        **else if** $s$ is terminal **then**
            **output** $\perp$
    **function** Env_eval_state($s$, $\mathcal{F}$)
        **output** $\phi_{\mathcal{G}}(s)$
    **function** Env_encode($s$)
        **output** encode_state($s$)

---

- These problems are intractable in the horizon for RL methods that are model-free using $\mathcal{G} = \mathrm{OP}(\mathcal{M})$ following Definition B.1.

- These problems are solved efficiently in the horizon by an RL method that takes the family $\mathcal{M}$ as input.

### B.2   An extension of the formalism

We extend their definition of a model-free method with an interface. Similarly to the first interface we defined in Algorithm 2, this interface uses the encoder-decoder (Algorithm 1) to hide the content of states while still allowing their manipulation. Thus, as for the previously defined interface, the methods can generate transitions and do local planning, such as tree-search.

**Definition B.2.** A *black-box dynamics RL method* interacts only with the interface defined in Algorithm 4 linked with the RL problem to solve and $\mathcal{G} = \mathrm{OP}(\mathcal{M})$ for $\mathcal{M}$ a given set of problems containing the problem to solve.

### B.3   New result

We derive a new version of our theorem in this formalism. The structure of the result is the same: we provide a family of RL problems which are both hard to solve for the class of RL methods that satisfies our definition, but is efficiently solved by another algorithm.

First, we define Algorithm 5. This method is similar to Algorithm 3 with two differences. One, the goal-state can be anywhere in the trajectory, while Algorithm 3 required it to be one of the final states. Two, the reward that decides the goal-state is not gained when entering the state, but when leaving it.

These changes are made to adapt to the new family of problems that will be used in the proof. Notice that Algorithms 3 and 5 both simply assign a state that appeared next to a high reward as a goal to reach.

**Theorem B.3.** *There exists a family $\mathcal{M}$ of RL problems such that*

1. *For any problem in the family and $\delta \in (0, 1)$, with probability at least $1 - \delta$ (over the sampled trajectories), Algorithm 5 outputs an optimal policy with a number of samples and number of operations upper bounded by a polynomial in horizon $H$ and $1/\delta$.*

---

**Algorithm 5** A second goal-conditioned algorithm.

---

**Parameters:** $N$: number of samples, $\alpha$: parameter of the learning algorithm (number of active features)

**Input:** $P$: the operator corresponding to the RL problem to solve

$\quad D \leftarrow \{\}$

$\quad$**for** $i \leftarrow 1, \dots N$ **do**

$\quad\quad D \leftarrow D \cup \{(s_0, a_0, r_0, s_1, a_1, \dots, s_H) \sim P^{\pi^U}\}$

$\quad g_T \leftarrow \arg\max_{s_T \in \mathcal{S}} r_T \quad$ s.t. $\quad (\dots, s_T, a_T, r_T, \dots) \in D$

$\quad$**for** $t \in \{0, \dots, H-1\}$ **do**

$\quad\quad D_{GC}^t \leftarrow \{((s_t, s_T), a_t) \,|\, (\dots, s_t, a_t, \dots, s_T, \dots) \in D\}$

$\quad\quad f^t \leftarrow$ the result of the optimization program 1 with dataset $D_{GC}^t$ and parameter $\alpha$.

$\quad$**output** $\pi(a|s_t) = f^t(a|s_t, g_T)$

---

2. *For any algorithm satisfying Definition B.1 with the family $\mathcal{M}$ and using $o(2^H)$ calls to the interface (Algorithm 4), there exists a problem in the family for which it outputs a suboptimal policy with probability at least $1/3$.*

*Proof.* The family of RL problems we construct is the same as for the proof of Theorem 4.1 except in the last steps of the dynamics. An example of a problem in the family is represented in Figure 4.

We describe the changes in comparison with the previous family of RL problems.

The rewards before entering the final states of the previous dynamics are all set to zero. A new step is added at the end of the dynamics where all states and actions lead to the same state full of zeros. The unique state that got a reward in the previous dynamics, leads now to a unitary reward in this last transition (the other states still lead to no reward).

For the first claim, the Algorithm 5 solves efficiently in the horizon the RL problems in the family, since the proof of Theorem 4.1 applies straightforwardly in this case. With high probability, the same goal state will be selected, and the same policy will then be learned.

We now prove the second claim. The left-hand dynamics is independent of $b$ the hidden binary word defining the RL problem. Thus, the only information the method will receive is from the right-hand dynamics. This dynamics is the same as the problems in the work of Sun et al. (2019) and their proof applies.

We repeat the elements of the proof here for completeness.

For an algorithm satisfying Definition B.2, the possible right-hand dynamics are equivalent to the family of problems defined in Proposition A.8 with a set of $2^{H-3}$ actions, and the proposition implies exactly the claim to prove.

We prove the equivalence. First, we note that any states encountered before the $t = H-2$ time step provide no information on the hidden binary word $b$. The same holds for the last state. Such that only the states at time step $t = H-1$ are relevant to discover $b$.

Let us describe $\phi_{\mathcal{G}}$ with $\mathcal{G} = \mathrm{OP}(\mathcal{M})$, applied on these states. For a fixed horizon $H$, there exists as many different dynamics in $\mathcal{M}$, as the number of possible binary words $b$. Each of these dynamics entails both an optimal Q-value function and an optimal policy function. The outputs of policy functions contain no information since the dynamics are independent of the action at time step $t = H-1$. The outputs of Q-value functions will evaluate to zero for all couples of states and actions at time step $t = H-1$, except for the unique state leading to a reward, for which they will all evaluate to one.

Such that the output of $\phi_{\mathcal{G}}(s)$ will always be only zeros for all suboptimal states encountered at time step $t = H-2$, and only be different for the unique rewarding state. All the suboptimal states will thus be indistinguishable for a method satisfying Definition B.2. This is equivalent to the armed bandit problem described in Proposition A.8. $\qquad\square$

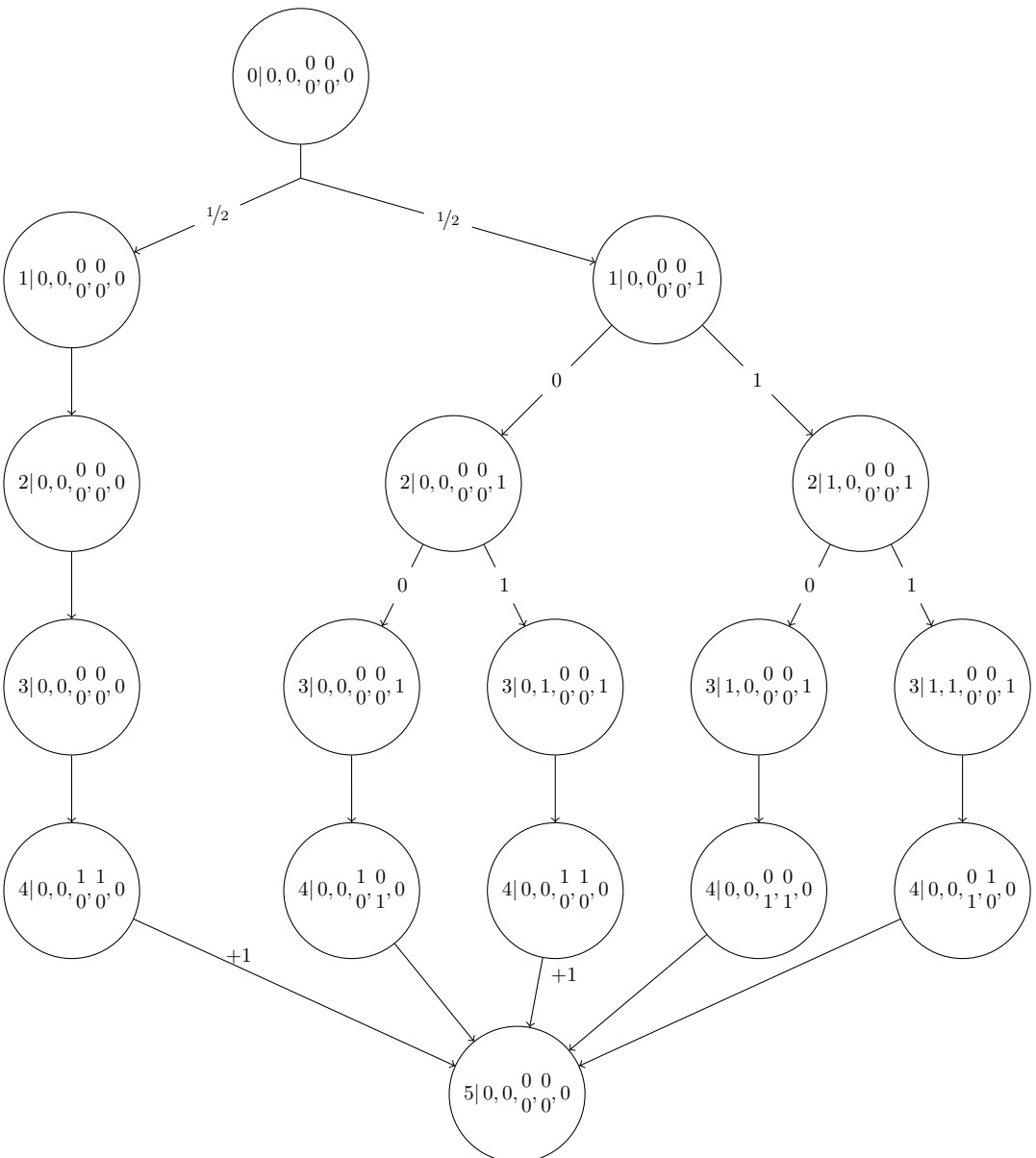

Figure 4: Example of the family of RL problems for horizon $H = 5$ and hidden binary word $b = 01$.

## B.4    Discussion

The result obtained in this appendix is very similar to the one presented in the main text. This new parallel version of our theoretical result offers a new trade-off between the assumption made and the conclusion. We explain this trade-off.

We compare Definition B.2 and linked Theorem B.3 of this appendix section with Definition 3.2 and linked Theorem 4.1 presented in the main text.

As we explain and show in the main text and Appendix D, the Definition 3.2 covers many common RL methods.

In contrast, the original assumption of Sun et al. (2019) (Definition B.1 and the discussion below it) and our extension in Definition B.2, pose conditions that do not seem to fit straightforwardly classical RL methods.

These assumptions rely on looking states through a set of functions defined by $\text{OP}(\mathcal{M})$, the set of optimal Q-functions and policy functions for a family $\mathcal{M}$ of RL problems. However, classical algorithms will use and evaluate encountered states using functions that cannot be linked to any arbitrary set of problems $\mathcal{M}$. For example, an algorithm could evaluate states with linear value functions but $\text{OP}(\mathcal{M})$ might not contain these depending on the family $\mathcal{M}$ chosen.

Despite this problem, the assumption of Sun et al. (2019) and of this appendix still seems relevant. It seems counter-intuitive that a Q-learning methods needs to rely on sub-optimal Q-value functions to solve a problem.

If one accepts this argument, there is a gain on what can be proved. We believe that the family of RL problems used in the proof of Theorem B.3 are easier to generalize than the ones in the proof of Theorem 4.1. The family of problems defined in this appendix sets the rewards after the states of interest not only just before, as pictured in Figure 4. This allows a possible generalization, which is to extend the dynamics beyond the current last states and delay the gain in rewards. Thus, the goal-state used in our proof could become an intermediary sub-goal in a larger dynamics.

## C   Comparison with the work of Sun et al. (2019)

The result in the main text and its parallel version in the previous section differ in two main ways from the result of Sun et al. (2019).

First, we redefine and broaden the class of methods with a limitation. Instead of proving a limitation on model-free methods only, we broaden the result of intractability to some methods that are usually considered model-based (such as AlphaZero). We obtain this by analyzing the computational complexity instead of the statistical complexity of the methods and by using an interface with an encoder-decoder of states. The interface allows more flexibility in the interactions between the method and the problem at hand, allowing us to abstract a larger class of methods. In particular, the interface allows to generate different potential future trajectories from a given state, something which is essential for several planning algorithms with a known or learned model of the dynamics.

Our second contribution with respect to the result of Sun et al. (2019) is that our family of problems can be efficiently solved without strong specific priors about it. In contrast, their result crucially relies on an a priori known goal-state to reach. They encode this knowledge into a model-based method to its advantage and show that their model-free methods cannot leverage that prior information. In general, such knowledge cannot be assumed to be known in practice and impairs the practical applicability and generality of their claim.

Whereas in our result, no goal-state needs to be known a priori but it is discovered autonomously from obtained rewards. Thus, our result applies without needing to know and engineer strong a priori knowledge into an algorithm to get an advantage.

This leads to a new claim specific to our work: numerous classical RL methods of the literature have a fundamental limitation on problems that should be solvable by an efficient *general-purpose* RL method.

## D   Classical RL methods implemented with the interface

In this Appendix, we reformulate common RL methods to use the interface defined in 2 instead of directly interacting with the RL problem dynamics. We thus demonstrate that these methods can be cast in the conditions of Definition 3.2 and suffer an upper bound on their performance provided by Theorem 4.1.

To clarify the presentation, we use methods as simplified as possible. For example, we only work with one-step RL methods, which use only one-step transition, $(s, a, r, s')$, instead of longer sequences to make updates. The presented formalism can however be extended to include these multi-step methods. Our results are robust to such changes.

Neural network initialization refers to the initialization of a multilayer perceptron (MLP), a classical neural network architecture that composes iteratively linear operators and non-linear point-wise activation functions. We keep implicit the input dimension, output dimension, and number of layers with their respective numbers of hidden units.

In Algorithm 6, we define the fitted Q-iteration algorithm with Deep Learning, a common variant of model-free deep Q-learning methods Riedmiller (2005). The algorithm trains at each iteration a new Q-function based on the Bellman equation on a new dataset and the previously trained $Q$-function.

In Algorithm 7, we translate the original algorithm to use the interface defined in Algorithm 2. As explained in the main text, calls to draw trajectories from the RL problem are replaced by calls to Env_step. Instead of directly manipulating states and updating the neural networks, the call to Env_step uses function $\mathcal{F}$ to do these computations. The function $\mathcal{F}$ is used to learn and evaluate the neural networks (Q-value functions).

Following the argument formalized in Appendix E about the symmetries existing in neural networks training by gradient descent, the functions defined in $\mathcal{F}$ practically satisfy the symmetry condition of Definition 3.2.

We also implement an Actor-Critic method, defined in Algorithm 8, with the interface in Algorithm 9. This demonstrates the use of value and policy functions.

In Algorithm 10, we implement a local planning tree search procedure relying on the capacity of the interface to generate transitions at any given state.

It is possible to combine tree search and model-free methods to fit methods that leverage feedback from results of tree search procedures to train neural networks Feinberg et al. (2018), then possibly improve the tree search procedure with these neural networks in a virtuous cycle Anthony et al. (2017); Silver et al. (2017).

We fit the AlphaZero algorithm to our Definition 3.2 in Algorithms 13 and 14, the initial algorithm is described in Algorithms 11 and 12.

The original algorithm was described for 2 players zero-sum deterministic games with the use of self-play, but is straightforwardly appliable to deterministic RL problems. In addition, we note the following minor modifications with respect to the original AlphaZero described in Silver et al. (2016; 2017):

- We evaluate the neural networks at each new node in the search-tree that is created, and not only at each leaf-node expanded. This incurs only a fixed factor more computations.

- Instead of using the final return of a full trajectory to learn the value function, we use one-step temporal differences. We do this because our Definition 3.2 does not support multi-steps temporal difference in the presented form. We note that multi-steps methods could be added to the formalism but at the cost of readability.

- To simplify the algorithm, we describe an online training version. We apply stochastic gradient descent directly with each new data sample, and do not construct datasets to leverage experience replay.

## E   Symmetries in neural networks learning

We provide here a formal argument that classical neural network training procedures satisfy the condition in Definition 3.2 put on function $\mathcal{F}$.

We prove that the distribution of outputted functions by a gradient descent procedure respects such a symmetry.

In Definition E.1, we construct a symmetry condition that will help us to state and to prove our result. In Theorem E.2, we state the main result of this section on the symmetry of the learning procedure. Corollary E.3 restate the Theorem, but in a form which can directly be compared with the condition on $\mathcal{F}$ stated in Definition 3.2.

---

**Algorithm 6** Fitted Q-iteration (with $\epsilon$-greedy exploration)

---

**Parameters:** $H$: horizon of the MDP, $\mathcal{S}$: state space, $\mathcal{A}$: action space, $K$: number of iterations, $\eta > 0$: scalar factor for gradient descent, $I$: number of samples by iteration (divisible by $H$), $\epsilon$ exploration parameter

**Input:** $P$: the operator corresponding to the MDP to solve

   Initialize a neural network $Q_\theta : \mathcal{S} \to \mathbb{R}^{|A|}$

   $\pi_\epsilon(a \,|\, s, Q) \leftarrow (1 - \epsilon)\mathbf{1}(a = \arg\max_{a \in A} Q(s, a)) + \epsilon/2\,\mathbf{1}(a = 0) + \epsilon/2\,\mathbf{1}(a = 1)$

   **for** $k \leftarrow 1, \dots, K$ **do**

      $\bar{Q} \leftarrow Q_\theta$

      Initialize a neural network $Q_\theta$

      **for** $i \in 1, \dots, I/H$ **do**

         $s \leftarrow P_0$

         **for** $t \leftarrow 0 \dots H - 1$ **do**

            $a \sim \pi_\epsilon(a \,|\, s, \bar{Q})$

            $r, s' \leftarrow P_{\text{dyn}}(r, s' \,|\, s, a)$

            $\theta \leftarrow \theta - \eta \nabla_\theta \left[ Q_\theta(s, a) - r - \max_{a \in \mathcal{A}} \bar{Q}(s', a) \right]^2$

            $s \leftarrow s'$

   $\pi(a \,|\, s) \leftarrow \pi_0(a \,|\, s, Q_\theta)$

   **output** $\pi$

---

**Algorithm 7** Fitted Q-iteration implemented with the interface defined in Algorithm 2. The function chunk$(D, I)$ partitions the data sequence $D$ into contiguous subsequences of length $I$ until there is less than $I$ elements left which are put in the last subsequence.

---

**Parameters:** $H$: horizon of the MDP, $\mathcal{S}$: state space, $\mathcal{A}$: action space, $K$: number of iterations, $\eta > 0$: scalar factor for gradient descent, $I$: number of samples by iteration, $\epsilon$ exploration parameter

**Input:** Env: the MDP interface with its methods defined in Algorithm 2

   $\pi_\epsilon(a \,|\, Q_s) \leftarrow (1 - \epsilon)\mathbf{1}(a = \arg\max_{a \in A} Q_s(a)) + \epsilon/2\,\mathbf{1}(a = 0) + \epsilon/2\,\mathbf{1}(a = 1)$

   **function** learnQ$(s, D)$

      Initialize a neural network $Q_\theta : \mathcal{S} \to \mathbb{R}^{|\mathcal{A}|}$

      $D_1, \dots, D_{n-1}, D_n \leftarrow \text{chunk}(D, I)$

      **for** $(s_d, a_d, r_d, Q_{s'_d}) \leftarrow D_{n-1}$ **do**         ▷ Only use $D_{n-1}$ with the most recently fixed Q-values

         $\theta \leftarrow \theta - \eta \nabla_\theta \left[ Q_\theta(s_d, a_d) - r_d - \max_{a \in \mathcal{A}} Q_{s'_d}(a) \right]^2$

      **output** $[Q_\theta(s, a)]_{a \in \mathcal{A}}$

   $\mathcal{F} \leftarrow \text{learnQ}$

   **for** $k \leftarrow 1, \dots, K \cdot I/H$ **do**

      $\bar{s}, Q_s \leftarrow \text{Env\_init}(\mathcal{F})$

      **for** $t \leftarrow 0, \dots, H - 1$ **do**

         $a \sim \pi_\epsilon(a \,|\, Q_s)$

         $r, \bar{s}, Q_s \leftarrow \text{Env\_step}(\bar{s}, a, \mathcal{F})$

   $\pi(a \,|\, s) \leftarrow \mathbf{1}(a = \arg\max_{a \in A} \text{Env\_eval\_state}(s, \text{learnQ}))$

   **output** $\pi$

---

For the clarity of the exposition, we will abusively refer to the probability of a function instead of the probability of events in the related implicit $\sigma$-algebra.

**Definition E.1.** Symmetry condition.

Let $L$ be a function from a sequence of data points in $(\mathbb{R}^n \times \mathbb{R})^N$ for some natural number $N$, to a set of distributions over functions $\Delta(\{\mathbb{R}^n \to \mathbb{R}\})$.

For any sequence $d = ((x_0, y_0), \dots)$ of data points and for any permutation of the coordinates $p$ over $\mathbb{R}^n$ (Definition 3.1), we define $d_p = ((p(x_0), y_0), (p(x_1), y_1), \dots)$.

---

**Algorithm 8** Deep Policy Gradient with Value function learning (Actor-Critic).

---

**Parameters:** $H$: the horizon of the MDP, $\mathcal{S}$: state space, $\mathcal{A}$: action space, $K$: number of sampled trajectories, $\eta > 0$: scalar factor for gradient descent, $I$: number of iterations by initialization of the value function (should be divisible by $H$)

**Input:** $P$: the operator corresponding to the MDP to solve

    Initialize a neural network $\pi_{\theta_1} : \mathcal{S} \to \Delta(\mathcal{A})$
    Initialize a neural network $V_{\theta_2} : \mathcal{S} \to \mathbb{R}$
    **for** $k \leftarrow 1 \ldots K$ **do**
        **if** $k \cdot H \mod I = 0$ **then**
            $\bar{V} \leftarrow V_{\theta_2}$
            Initialize a neural network $V_{\theta_2} : \mathcal{S} \to \mathbb{R}$
        $s \leftarrow P_0$
        **for** $t \leftarrow 0, \ldots, H - 1$ **do**
            $a \sim \pi_{\theta_1}(a \,|\, s)$
            $r, s' \leftarrow P_{\mathrm{dyn}}(r, s' \,|\, s, a)$
            $\theta_1 \leftarrow \theta_1 + \eta(r + \bar{V}(s'))\nabla_{\theta_1} \log \pi_{\theta_1}(a \,|\, s)$
            $\theta_2 \leftarrow \theta_2 - \eta \nabla_{\theta_2} \left[V_{\theta_2}(s) - r - \bar{V}(s')\right]^2$
            $s \leftarrow s'$
    **output** $\pi_{\theta_1}$

---

**Algorithm 9** Deep Policy Gradient with Value function learning (Actor-Critic) implemented with the interface defined in Algorithm 2. The function $\mathrm{chunk}(D, I)$ partitions the data sequence $D$ into contiguous subsequences of length $I$ until there is less than $I$ elements left which are put in the last subsequence. We simplify the learning of $V$, the value function, following the discussion in Section E.

---

**Parameters:** $H$: horizon of the MDP, $\mathcal{S}$: state space, $\mathcal{A}$: action space, $K$: number of sampled trajectories, $\eta > 0$: scalar factor for gradient descent, $I$: number of iterations by initialization of the value function

**Input:** Env: the MDP interface with its methods defined in Algorithm 2

    **function** learningVand$\pi(s, D)$
        Initialize a neural network $V_{\theta_2} : \mathcal{S} \to \mathbb{R}$
        $D_1, \ldots, D_{n-1}, D_n \leftarrow \mathrm{chunk}(D, I)$
        **for** $(s, a, r, (V_{s'}, \pi_{s'})) \leftarrow D_{n-1}$ **do**
            $\theta_2 \leftarrow \theta_2 - \eta \nabla_{\theta_2} \left[V_{\theta_2}(s) - r - V_{s'}\right]^2$
        Initialize a neural network $\pi_{\theta_1} : \mathcal{S} \to \mathbb{R}^{|A|} (= \Delta(\mathcal{A}))$
        **for** $(s, (a, r, (V_{s'}, \pi_{s'}))) \leftarrow D$ **do**
            $\theta_1 \leftarrow \theta_1 + \eta(r + V_{s'})\nabla_{\theta_1} \log \pi_{\theta_1}(a \,|\, s)$
        **output** $(V_{\theta_2}(s), \pi_{\theta_1}(s))$
    $\mathcal{F} \leftarrow$ learningVand$\pi$
    **for** $k \leftarrow 1 \ldots K$ **do**
        $\bar{s}, (V_s, \pi_s) \leftarrow \mathrm{Env\_init}(\mathcal{F})$
        **for** $t \leftarrow 0 \ldots H - 1$ **do**
            $a \sim \pi_s(a)$
            $r, \bar{s}, (V_s, \pi_s) \leftarrow \mathrm{Env\_step}(\bar{s}, a, \mathcal{F})$
    $\pi(a \,|\, s) \leftarrow [(V_s, \pi_s) \leftarrow \mathrm{Env\_eval\_state}(s, \mathcal{F}); \quad$ **output** $\pi_s(a)]$
    **output** $\pi$

---

The function $L$ satisfies the symmetry condition if, for any permutation of the coordinates $p$, we have $L(d) = L(d_p) \circ p^{-1}$. In other words, for any function $h : \mathbb{R}^n \to \mathbb{R}$, the output with $d$ must give the same probability on $h$ as the output with $d_p$ on $h \circ p$.

---

**Algorithm 10** Tree-search implementation using the interface defined in Algorithm 2. This algorithm can be combined with the other methods presented, to use value function estimates instead of rewards for example.

---

**Parameters:** $H$: horizon of the MDP, $\mathcal{A}$: action space, $H_S > 0$: search horizon of the tree

**Input:** Env: the MDP interface with its methods defined in Algorithm 2

> **function** TREE_SEARCH($\bar{s}$, $h$)
>> **if** $h = 0$ **then**
>>> **output** _ , 0
>>
>> **for** $a \in \mathcal{A}$ **do**
>>> $r, \bar{s}', \_ \leftarrow$ Env_step($\bar{s}$, $a$, _)
>>> **if** $r = 1$ **then**
>>>> **output** $a$, 1
>>>
>>> **else**
>>>> $a', R \leftarrow$ Tree_search($\bar{s}'$, $h - 1$)
>>>> **if** $R = 1$ **then**
>>>>> **output** $a$, 1
>>
>> **output** _ , 0
>
> **function** TREE_SEARCH($s_t$)
>> $a, r \leftarrow$ Tree_search(Env_encode($s_t$), $\min\{H_S, H - t\}$)
>> **output** $a$
>
> $\pi(a\,|\,s_t) \leftarrow$ Tree_search($s_t$)
> **output** $\pi$

---

**Theorem E.2.** *The output of the randomized Algorithm 15, which produces a learned function given a dataset, follows a distribution defined by the dataset. The function between the dataset and the distribution of learned functions respects the symmetry condition.*

*Proof.* We prove by induction that for any permutation of the coordinates $p$ both the process that constructs the linear operator represented by $W$ and the processes that produce all the other functions of the input in the neural networks (output, gradients) satisfy the symmetry condition.

Let $P$ be the permutation matrix corresponding to $p$.

At initialization, the symmetry condition is satisfied everywhere. The matrix $W$ is as probable as $WP^{-1}$ and is independent of the dataset. All the other functions computed in the neural network depend on the input only through the linear operator that satisfies the condition, such that they also satisfy the symmetry condition.

Now we look at the updates. Let $g$ be the gradient at the output of the linear operator for some input $x$. Then the update of $W$ is

$$W \leftarrow W - \eta g x^T. \tag{8}$$

On the transformed dataset, with $WP^{-1}$ the matrix of the linear operator, $Px$ the new input, and $h$ the new gradient, the update becomes

$$WP^{-1} \leftarrow WP^{-1} - \eta h(Px)^T, \tag{9}$$

which reduces to $(W - \eta h x^T)P^{-1}$. By the inductive hypothesis, we know that the processes that produce $h$ respect the symmetry condition, thus the gradient $h$ is sampled from the same distribution as $g$. Also, by the inductive hypothesis, the previous $WP^{-1}$ is as probable as the previous $W$ under their respective distributions. Consequently, the updated $W$ is as probable as the updated $WP^{-1}$.

Similarly to the initialization, the processes that produce the rest of the network functions of the input respect the symmetry condition because they depend on the input only through the output of the linear operator since the start of the algorithm.

$\square$

---

**Algorithm 11** AlphaZero planning functions.

---

**Parameters:** $H$: the horizon of the MDP, $\mathcal{S}$: state space, $\mathcal{A}$: action space

**Input:** $P$: the operator corresponding to the deterministic MDP to solve

  Node: N, W, Q, P, v, s, children, parent, parent_action, leaf, h

  **function** PLANNING($(s, h, \pi_{\theta_1}, V_{\theta_2}, K, c, \tau)$)

    nodes $\leftarrow$ []

    nodes.append(

      Node($0^{\mathcal{A}}, 0^{\mathcal{A}}, 0^{\mathcal{A}}$,null,null,$s$,null,null,null,True,h))

    **for** $k \leftarrow 1 \ldots K$ **do**

      last_node$\leftarrow$Select(nodes[0], $c$)

      nodes.concatenate(Expand(last_node, $\pi_{\theta_1}$, $V_{\theta_2}$))

      Backup(last_node.parent, last_node.$v$)

    **output** root_node.N[a]$^{1/\tau}$ / $\sum_{a \in \mathcal{A}}$root_node.N[a]$^{1/\tau}$

  **function** SELECT(node, $c$)

    **if** node.leaf or node.h equal $H$ **then**

      **output** node

    $a \leftarrow \arg\max_{a \in \mathcal{A}}$curr_node.Q[a] $+ c$ node.P[a]$\frac{\sqrt{\sum_{b \in \mathcal{A}} \text{node.N}[b]}}{1+\text{node.N}[a]}$

    **output** Select(node.children[a])

  **function** EXPAND(node, $\pi_{\theta_1}$, $V_{\theta_2}$)

    node.leaf$\leftarrow$ False

    **if** node.h equal $H$ **then**

      stop

    node.children $\leftarrow$ []

    **for** $a \in \mathcal{A}$ **do**

      r,s'$\leftarrow P_{\text{dyn}}(r, s'|\text{node}.s, a)$

      v'$\leftarrow V_{\theta_2}$(s')

      node.children.append(Node($0^{\mathcal{A}}, 0^{\mathcal{A}}, 0^{\mathcal{A}}$,null,v',s',null,node,$a$,True,node.h+1))

    node.P$\leftarrow \pi_{\theta_1}$(node.s)

    **output** node.children

  **function** BACKUP(node, $a$, $v$)

    **if** node is null **then**

      **stop**

    node.N[$a$] $\leftarrow$ node.N[$a$]+1

    node.W[$a$] $\leftarrow$ node.W[$a$] + v

    node.Q[$a$] $\leftarrow$ node.W[$a$]/node.N[$a$]

    Backup(node.parent, node.parent_action, $v$)

---

**Corollary E.3.** *The expectation of the distribution of the output of neural networks applied on some input $z \in \mathbb{R}^n$ produced by randomised Algorithm 15 for some sequence of points d can be written as $f(z, ((x_0, y_0), \ldots))$ for some function f that is invariant to a permutation of the coordinates applied to z and $x_0, \ldots$*

*Proof.* By construction, $f$ exists and is simply the function representing the process producing the distribution of learned functions composed with the expectation of their evaluations. Since the distribution of learned functions respects the symmetry condition by Theorem E.2, we know that for any permutation of the coordinates $p$, we have $\mathbb{E}_{\mathbf{nn} \sim \mathbf{Alg}(d)}[\mathbf{nn}(z)] = \mathbb{E}_{\mathbf{nn} \sim \mathbf{Alg}(d_p)}[\mathbf{nn}(p(z))]$ where $\mathbf{nn}$ is the neural network trained by the algorithm $\mathbf{Alg}$. $\square$

---

**Algorithm 12** AlphaZero.

---

**Parameters:** $H$: the horizon of the MDP, $\mathcal{S}$: state space, $\mathcal{A}$: action space, $I$: number of iterations of the procedure, $K$: number of expansions in the search, $\eta > 0$: scalar factor for gradient descent, $\tau$ temperature for action selection, $c$ exploration parameter of the search

**Input:** $P$: the operator corresponding to the deterministic MDP to solve

    Initialize a neural network $\pi_{\theta_1} : \mathcal{S} \to \Delta(\mathcal{A})$
    Initialize a neural network $V_{\theta_2} : \mathcal{S} \to \mathbb{R}$
    **for** $i \leftarrow 1 \dots I$ **do**
        $s \leftarrow P_0$
        **for** $h \leftarrow 0, \dots, H-1$ **do**
            $p \leftarrow \text{planning}(s, h, \pi_{\theta_1}, V_{\theta_2}, K, c, \tau)$
            $\theta_1 \leftarrow \theta_1 + \eta \nabla_{\theta_1} \sum_{a \in \mathcal{A}} p(a) \log \pi_{\theta_1}(a \,|\, s)$
            $a \sim p$
            $r, s' \leftarrow P_{\text{dyn}}(r, s' \,|\, s, a)$
            $\theta_2 \leftarrow \theta_2 - \eta \nabla_{\theta_2} \left[ V_{\theta_2}(s) - r - \bar{V}(s') \right]^2$
            $s \leftarrow s'$
    $\pi(a \,|\, s) \leftarrow [p \leftarrow \text{planning}(s, s.h, \pi_{\theta_1}, V_{\theta_2}, K, c, \tau); \quad \textbf{output } p(a)]$
    **output** $\pi$

---

## F   A planning method

In this appendix, we describe the planning method introduced in the numerical experiments section of the main text.

The method follows these steps:

1. Sample an initial dataset of trajectories using a uniform policy.

2. Learn a model of the forward dynamics by fitting the dataset with Decisions Trees auto-regressively assuming a deterministic transition dynamics. If a learned model of the dynamics already existed from previous iterations combine the previously learned and new Decision Trees predicting rewards to produce a pessimistic prediction of obtained rewards. In other words, the method combines additively the constraints predicting zero rewards in the trees.

3. Sample trajectories following a policy that plans each step using the learned model of the dynamics. The planning is done by solving a Mixed-Integer Linear Programming (MILP) optimization problem. The main variables in this program represent the future trajectory (states, actions, rewards), there are also auxiliary variables that help encode the constraints coming from the learned dynamics. The constraints of the MILP are given by: the current state and the model of the dynamics defined by Decision Trees. The objective is to maximize the total return of the trajectory. To solve the MILP, we use the off-the-shelf solver Gurobi Gurobi Optimization, LLC (2023).

4. If the sampled trajectories reach reach a threshold in average returns, the method outputs the policy that plans each action by solving a MILP encoding the learned dynamics (as in the last step). Else, if the performance is not sufficient, add the trajectories to the dataset and the method goes back to step 2.

An algorithmic description of the method that represents the loops and branching statements more clearly is provided in Algorithm 16.

The method requires a sufficiently deterministic dynamics. The dynamics should also be simple enough for Decision Trees to be able to efficiently model it with a dataset of trajectories produced with a uniform policy, followed by a low exploration with several trajectories that are promising according to the dynamics model. This low exploration comes from the compromise of choosing a pessimistic prediction of the rewards.

---

**Algorithm 13** AlphaZero planning functions implemented with the interface defined in Algorithm 2, fitting Definition 3.2.

---

**Parameters:** $H$: the horizon of the MDP, $\mathcal{S}$: state space, $\mathcal{A}$: action space

**Input:** Env: the MDP interface defined in Algorithm 2

   Node: N, W, Q, P, v, $\bar{s}$, children, parent, parent_action, leaf, h

   **function** PLANNING($\bar{s}, h, \mathcal{F}, K, c, \tau$)

      nodes ← []

      $(v_s, \pi_s, p)$ ← Env_eval_state($s, \mathcal{F}$)

      nodes.append(Node($0^{\mathcal{A}}, 0^{\mathcal{A}}, 0^{\mathcal{A}}, \pi_s$,null,$\bar{s}$,null,null,null,True,h))

      **for** $k ← 1 \ldots K$ **do**

         last_node←Select(nodes[0], $c$)

         nodes.concatenate(Expand(last_node, $\mathcal{F}$))

         Backup(last_node.parent, last_node.$v$)

      **output** root_node.N[a]$^{1/\tau}/\sum_{a\in\mathcal{A}}$root_node.N[a]$^{1/\tau}$

   **function** SELECT(node, $c$)

      **if** node.leaf or node.h equal $H$ **then**

         **output** node

      $a ← \arg\max_{a\in\mathcal{A}}$curr_node.Q[a] $+ c$ node.P[a]$\frac{\sqrt{\sum_{b\in\mathcal{A}}\text{node.N}[b]}}{1+\text{node.N}[a]}$

      **output** Select(node.children[a])

   **function** EXPAND(node, $\mathcal{F}$)

      node.leaf← False

      **if** node.h equal $H$ **then**

         stop

      node.children ← []

      **for** $a \in \mathcal{A}$ **do**

         $r, \bar{s}', (v_{s'}, \pi_{s'}, p_s)$ ← Env_step(node.$\bar{s}$, $a$, $\mathcal{F}$, append_to_data = False)

         node.children.append(Node($0^{\mathcal{A}}, 0^{\mathcal{A}}, 0^{\mathcal{A}}, \pi_{s'}, v_{s'}, \bar{s}'$,null,node,$a$,True,node.h+1))

      **output** node.children

   **function** BACKUP(node, $a$, $v$)

      **if** node is null **then**

         **stop**

      node.N[$a$] ← node.N[$a$]+1

      node.W[$a$] ← node.W[$a$] + v

      node.Q[$a$] ← node.W[$a$]/node.N[$a$]

      Backup(node.parent, node.parent_action, $v$)

---

As for Algorithm 3 this method makes several assumptions about the simplicity of the problems it solves. Similarly, those assumptions are not specific to the problems in this paper, but prevent the method from being a general-purpose RL method. The method only illustrates an algorithmic approach to RL on our problems and shows that these problems are easy to solve.

# G   Numerical experiments details

We describe the implementation of the different methods we test: a goal-conditioned method in Algorithm 17, fitted Q-iteration Riedmiller (2005) in Algorithm 18, Proximal Policy Optimization (PPO) Schulman et al. (2017), and AlphaZero Silver et al. (2017) in Algorithm 12. We refer to the previous section for details on the planning method introduced in the numerical experiments section of the main text.

We test all these algorithms with MLP neural networks. Any state in input to a neural network $s = (t, x) \in \mathcal{S} = \{(t, x) \in \{0, \ldots, H\} \times \mathbb{R}^n\}$ will be represented with a one-hot encoding of the time step $t$ concatenated to the real part $x$.

---

**Algorithm 14** AlphaZero implemented with the interface defined in Algorithm 2, fitting Definition 3.2.

---

**Parameters:** $H$: the horizon of the MDP, $\mathcal{S}$: state space, $\mathcal{A}$: action space, $I$: number of iterations of the procedure, $K$: number of expansions in the search, $\eta > 0$: scalar factor for gradient descent, $\tau$ temperature for action selection, $c$ exploration parameter of the search

**Input:** Env: the MDP interface with its methods defined in Algorithm 2

    **function** LEARNVAND$\pi(s, D \,|\, p)$

        Initialize a neural network $V_{\theta_2} : \mathcal{S} \to \mathbb{R}$

        **for** $(s, a, r, (V_{s'}, \pi_{s'}, p_{s'})) \leftarrow D$ **do**

            $\theta_2 \leftarrow \theta_2 - \eta \nabla_{\theta_2} \left[ V_{\theta_2}(s) - r - V_{s'} \right]^2$

        Initialize a neural network $\pi_{\theta_1} : \mathcal{S} \to \mathbb{R}^{|\mathcal{A}|}(= \Delta(\mathcal{A}))$

        **for** $(s, a, r, (V_{s'}, \pi_{s'}, p_s)) \leftarrow D$ **do**

            $\theta_1 \leftarrow \theta_1 + \eta \nabla_{\theta_1} \sum_{a \in \mathcal{A}} p_s(a) \log \pi_{\theta_1}(a \,|\, s)$

        **output** $(V_{\theta_2}(s), \pi_{\theta_1}(s), p)$

  $p \leftarrow 0$

  $\mathcal{F} \leftarrow$ learnVand$\pi(., . \,|\, p)$

  **for** $i \leftarrow 1 \ldots I$ **do**

    $\bar{s}, (V_s, \pi_s, p_s) \leftarrow$ Env_init$(\mathcal{F})$

    **for** $h \leftarrow 0, \ldots, H - 1$ **do**

        $p \leftarrow$ planning$(\bar{s}, h, \mathcal{F}, K, c, \tau)$

        $a \sim p$

        $\mathcal{F} \leftarrow$ learnVand$\pi(., . \,|\, p)$

        $\_, \bar{s}, \_ \leftarrow$ Env_step$(\bar{s}, a, \mathcal{F})$

  $\pi(a \,|\, s) \leftarrow [p \leftarrow$ planning$($Env_encode$(s), s.h,$ learnVand$\pi(., . \,|\, 0), K, c, \tau);$   **output** $p(a)]$

  **output** $\pi$

---

---

**Algorithm 15** Neural network training for a neural network with parameters $\theta = (W \in \mathbb{R}^{m \times n}, \theta' \in \mathbb{R}^k)$ for some $n, m, k \in \mathbb{N}$, which is interpreted in $\mathrm{nn}_\theta(x) = q_{\theta'}(Wx)$ for $q$ that outputs a real and is smooth w.r.t. $\theta'$ and its input.

---

**Parameters:** $\eta > 0$: a scalar factor for gradient descent, loss: $\mathbb{R} \times \mathbb{R} \to \mathbb{R}$: a smooth loss function w.r.t its inputs

**Input:** $D$: a sequence of couples $(x \in \mathbb{R}^n, y)$

  Initialize $\theta'$ and matrix $W$. The elements of the matrix $W$ are initialized i.i.d.

  **for** $(x, y) \in D$ **do**

    $\theta \leftarrow \theta - \eta \nabla_\theta$loss$(q_{\theta'}(Wx), y)$

  **output** $\mathrm{nn}_{\theta = (W, \theta')}$

---

Some notations, the function CE : $\Delta(X) \times X \to \mathbb{R}$ (Cross-Entropy) computes $\mathrm{CE}(q, x) = -\log q(x)$ for some set $X$. The function clip$(x \in \mathbb{R}, a \in \mathbb{R}, b \in \mathbb{R})$ computes $\min\{\max\{x, a\}, b\}$. The method copy copies the object, it allows to fix its state.

The optimizations are performed with mini-batches estimation of the gradient, and AdamW, a regularized version of Adam Loshchilov & Hutter (2017). Hyper-parameter optimization was performed with Bayesian Optimization to maximize success probability on problems with horizon $H = 45$ for the goal-conditioned algorithm, $H = 12$ for the fitted Q-iteration algorithm, and $H = 17$ for the PPO algorithm. For AlphaZero, we performed coordinate ascent optimization to tune the local search parameters for $H = 15$.

The goal-conditioned algorithm was run with 2 hidden layers of 256 units each, parameter $I = 1000$, 30000 steps of AdamW with learning rate $2 \cdot 10^{-2}$, weight decay $10^{-3}$, and 100 batches.

The fitted Q-iteration algorithm was run with 2 hidden layers of 512 units each, for $K = 50$ iterations, with $I = 1000$ sampled trajectories by iteration, $\epsilon = 0.2$, 1000 steps of AdamW by iteration, learning rate $1 \cdot 10^{-3}$, weigh decay $10^{-2}$, each 1000 trajectories samples were divided into 20 batches.

---

**Algorithm 16** A planning algorithm.

---

**Parameters:** $N$: number of initial samples.

**Input:** $P$: the transition operator corresponding to the RL problem to solve.

    Create dataset $D$ by sampling $N$ trajectories from the uniform policy $P^{\pi^U}$.

    **function** planning($s$, DTs)

        Build a Mixed-Integer Linear Program (MILP) that maximizes the future rewards given the current state $s$ and the learned dynamics given by Decision Trees DTs.

        Solve the mathematical program using an off-the-shelf MILP solver.

        **output** a plan, consisting in a sequence of actions, states and rewards.

    **while** True **do**

        Fit autoregressively Decision Tree classifiers DTs on the trajectories in $D$. Accumulate previously trained Decision Trees for the reward predictions (pessimistic prediction).

        Draw trajectories with $P$ by computing at each step a plan with planning($s$, DTs) and using the planned next action.

        If the computed plans reach a sufficient performance threshold **break**.

        Add the sampled trajectories to $D$.

    **output** a policy that computes a plan at each step given the learned dynamics model: planning(., DTs)

---

The PPO algorithm was run with 2 hidden layers of 512 units each, for $K = 50$ iterations, with $I = 1000$ trajectories sampled by iterations, clipping parameter $\epsilon = 0.1$, $\beta = 10^{-3}$, 500 steps of AdamW by iteration, learning rate of $2 \cdot 10^{-4}$, weight decay $10^{-8}$ and 10 batches of the sampled trajectories.

The AlphaZero algorithm was run with 2 layers of 256 neurons each, AdamW with learning rate $2 \cdot 10^{-3}$ and weight decay $1 \cdot 10^{-3}$. The algorithm performed the following operation three times: sampling a dataset of 1000 trajectories, then optimize an actor-critic for 30.000 steps with AdamW. For the local search procedure, we fixed a budget of 50 search node per decision and tuned the $c$ and $\tau$ parameters (see Algorithm 12).

We measured that it takes less than two days with a single GPU to reproduce the results provided in the numerical section with only one test by experiment (pair of horizon and method).

---

**Algorithm 17** Neural goal-conditioned algorithm.

---

**Parameters:** $\mathcal{S}$: state space, $\mathcal{A}$: action space, $I$: number of samples

**Input:** $P$: the operator corresponding to the MDP to solve

    $D \leftarrow \{\}$

    **for** $i \leftarrow 1, \ldots I$ **do**

        $D \leftarrow D \cup \{(s_0, a_0, r_0, s_1, a_1, \ldots, s_H) \sim P^{\pi^U}\}$

    $D \leftarrow \{(s_t, a_t, s_H) | (\ldots, s_t, a_t, \ldots, s_H) \in D\}$.

    Initialize neural network $f_\theta : \mathcal{S} \times \mathcal{S} \to \Delta(\mathcal{A})$

    $g \leftarrow \arg\max_{s_H} r \quad \text{s.t.} \quad (\ldots, r, s_H) \in D$.

    $\theta \leftarrow \arg\min_\theta \frac{1}{|D|} \sum_{(s, a, s_H) \in D} \text{CE}(f_\theta([s, s_H]), a)$

    **output** $\pi(a|s) = f_\theta(a|s, g)$

---

---

**Algorithm 18** Fitted Q-iteration

---

**Parameters:** $H$: horizon of the MDP, $\mathcal{S}$: state-space, $\mathcal{A}$: action space, $K$: number of iterations, $I$: number of trajectory samples by iteration, $\epsilon$: exploration parameter

**Input:** $P$: the operator corresponding to the MDP to solve

$\quad \pi_\epsilon^Q(a\,|\,s) \leftarrow (1-\epsilon)\mathbf{1}(a = \arg\max_{a\in A} Q(s,a)) + \epsilon/2\,\mathbf{1}(a=0) + \epsilon/2\,\mathbf{1}(a=1)$

$\quad$ Initialize a neural network $Q_\theta : \mathcal{S} \to \mathbb{R}^{|\mathcal{A}|}$

$\quad D \leftarrow \{\}$

$\quad$ **for** $k \leftarrow 1, \ldots, K$ **do**

$\qquad D' \leftarrow \{\}$

$\qquad$ **for** $i \leftarrow 1, \ldots, I$ **do**

$\qquad\quad D' \leftarrow D' \cup \{(s_0, a_0, r_0, s_1, \ldots, s_H) \sim P^{\pi_\epsilon^{Q_\theta}}\}$

$\qquad D \leftarrow D \cup D\{(s,a,r,s')\,|\,(\ldots, s, a, r, s', \ldots) \in D\}$

$\qquad \bar{Q} \leftarrow \text{copy}(Q_\theta)$

$\qquad \theta \leftarrow \underset{\theta}{\arg\min}\, \frac{1}{|D|} \sum_{(s,a,r,s')\in D} \left[ Q_\theta(s,a) - r - \max_{a\in\mathcal{A}} \bar{Q}(s',a) \right]^2$

$\quad$ **output** $\pi_0^{Q_\theta}(a\,|\,s)$

---

**Algorithm 19** Proximal Policy Optimization

---

**Parameters:** $H$: horizon of the MDP, $\mathcal{S}$: state-space, $\mathcal{A}$: action space, $K$: number of iterations, $I$: number of trajectory samples by iteration, $\epsilon$: clipping parameter, $\beta$: weight of the entropy regularization

**Input:** $P$: the operator corresponding to the MDP to solve

$\quad$ **function** PPO$\_$loss$(s, a, R\,|\,\pi_{\theta_1}, \bar{\pi}, \bar{V})$

$\qquad A \leftarrow R - \bar{V}(s)$

$\qquad r_\pi \leftarrow \frac{\pi_{\theta_1}(a|s)}{\bar{\pi}(a|s)}$

$\qquad$ **output** $\min\{r_\pi A, \text{clip}(r_\pi, 1-\epsilon, 1+\epsilon)A\} + \beta \cdot \text{entropy}(\pi_{\theta_1}(.|s))$

$\quad$ Initialize a neural network $\pi_{\theta_1} : \mathcal{S} \to \Delta(\mathcal{A})$

$\quad$ Initialize a neural network $V_{\theta_2} : \mathcal{S} \to \mathbb{R}$

$\quad$ **for** $k \leftarrow 1, \ldots, K$ **do**

$\qquad D \leftarrow \{\}$

$\qquad$ **for** $i \leftarrow 1, \ldots, I$ **do**

$\qquad\quad D \leftarrow D \cup \{(s_0, a_0, r_0, s_1, \ldots, s_H) \sim P^{\pi_{\theta_1}}\}$

$\qquad D \leftarrow \{(s_t, a_t, \sum_{j=t}^{H-1} r_j)\,|\,(\ldots, s_t, a_t, r_t, \ldots) \in D\}$

$\qquad \bar{\pi} \leftarrow \text{copy}(\pi_{\theta_1})$

$\qquad \bar{V} \leftarrow \text{copy}(V_{\theta_2})$

$\qquad \theta_1 \leftarrow \underset{\theta_1}{\arg\max}\, \frac{1}{|D|} \sum_{(s,a,R)} \text{PPO$\_$loss}(s, a, R\,|\,\pi_{\theta_1}, \bar{\pi}, \bar{V})$

$\qquad \theta_2 \leftarrow \underset{\theta_2}{\arg\min}\, \frac{1}{|D|} \sum_{(s,a,R)\in D} (V_{\theta_2}(s) - R)^2$

$\quad$ **output** $\pi_{\theta_1}(a\,|\,s)$

---

