# OpenReview forum: "A limitation on black-box dynamics approaches to Reinforcement Learning"
_TMLR — Accepted by TMLR_

### Review · Reviewer_kyzM · 2024-09-20

**Summary Of Contributions:**

This paper formulates a novel class of RL methods, containing model-free and model-based RL methods. A limitation of this class is discussed in this paper. The authors provided experimental results to show that such limitations can be solved by using learned dynamics models.

Contributions and new knowledge of this paper are:
- formulating a class of black-box dynamics methods, which assumes that RL algorithms access environment dynamics only by an interface.
- identifying a limitation of this class of methods, and providing the corresponding solution.

**Audience:**

Yes

**Broader Impact Concerns:**

No concerns on the ethical implications.

**Claims And Evidence:**

Yes

**Requested Changes:**

Please refer to the Weaknesses part above.

**Strengths And Weaknesses:**

### Strengths
- This paper is well presented and organized.
- The problem is quite interesting, as understanding the limitations of RL methods will help in designing new RL algorithms.
- The paper presented the connections to the previous work clearly.

### Weaknesses
- RL generally assumes that the environment is Markovian, where the observed state contains all the necessary information to predict the future state. But in the proposed formulation, states are not directly provided to the RL methods but only partial information about the state is provided.  As a result, the environment are not Markovian for the RL method. In this case, common RL methods do not fit the proposed definition. I would be grateful if the authors could provide explanation about it.
- The authors argue that the identified limitation is applied to model-free RL methods and some model-based methods treating dynamics models as a black-box. The experimental results in Fig.3 only show that model-free methods fail to solve the problem but lack evaluations of the existing model-based methods. Can the authors explain why you don't consider the model-based baseline in the experiment?

---

> ### Author Response · Authors · 2024-09-25
> **Answer**
>
> Thanks for your review.
>
> - In the appendices, we implement several classical RL methods with our interface such that the implementations' behaviors are the same as the original classical RL methods applied to the (Markovian) problems without the interface. Thus, our analysis of the implementations with the interface extends to the original RL methods with the original (Markovian) problems.
>
> - For clarity, we can conduct this additional experiment with some tree-search procedure applied to the family of problems. We note that the result will only be illustrative, as the inefficiency of such a method is already proved in the theoretical section.

---

### Review · Reviewer_UYVb · 2024-11-04

**Summary Of Contributions:**

This paper highlights a class of RL problems which many popular RL methods fail to solve, while there exist fairly simple methods to solve it efficiently.

To state this finding, the authors define a class of algorithms which is neither solely model-free or model-based but contains those said to take a 'black-box' view of the dynamics of the underlying MDPs. This concept is formalised by introducing a class of algorithms which can be posed as interacting with the problem through an 'interface' which masks true state transitions. This includes many popular and notable algorithms. Particular attention is given to AlphaZero.

The most similar work is that of Sun et al. (2019), which also separates classes of algorithm in terms of performance, but the present work differs as it a) encompasses some model-based algorithms within the class which suffers the limitations, and b) the problems in which the limitation is encountered can be specified without strong prior assumptions.

The paper carefully derives the class suffering the limitation, and gives an existence-based proof of problems such that methods in the class can be out-performed. An empirical example shows that RL algorithms within the class are dramatically outperformed by a planning solver, especially as the problem horizon increases. The paper concludes by identifying methods for which the limitation is not present.

**Audience:**

Yes

**Claims And Evidence:**

Yes

**Requested Changes:**

As per discussions above, there are three additions I feel could be made to improve the paper:

1. Granting a bit more detail towards the end of p3 and in the discussion of Fig 2 (p9) to support the reader in understanding the encoding process, and in particular what this looks like for the problem in Fig 2. This is quite a key aspect for a more casual reader to understand and I found I needed to re-read several times to get to grips with things in both cases.
2. A discussion of the feasibility of theory explaining which problems generate the limitation, as an extension to the result that there exists a family of problems for which the limitation presents itself.
3. Comments in Appendix A or the main text to explain where the differences in the theoretical contribution there and in the 'similar proofs' of Sun et al. (2019) are.

**Strengths And Weaknesses:**

The class of methods to which the limitation applies is wide and thus the finding is likely to be of wide interest. The paper is generally written well in the outset, but could be more user friendly in some key aspects (around the encoding of states and construction of the problem which provides the intuition for the main theoretical result.

The empirical results validate the theoretical finding, and I was unable to find issue with the proof. I do feel that it would be nice to give some sense of how one might go about deciding if a given problem is of the class for which black-box algorithms fail, as I understand it, at the moment, the proof shows the existence of problems for which black-box algorithms are limited, but not general conditions under which problems create the limiting behaviour. I appreciate this may be challenging, and necessitate further work, but some reflection on the feasibility of this would be welcome.

The other point where I feel some additional clarity would be beneficial is around in which ways the proof of Theorem 4.1 in Appendix A is similar to the proofs of Sun et al. (2019) as stated. I appreciate that in Appendix B, an alternative analysis is presented which connects the two, but the main paper says that the proofs are similar to those of Sun et al. (2019) and it is not clear from Appendix A where (if anywhere) the theoretical novelty may lie.

---

> ### Author Response · Authors · 2024-11-09
> **Answer**
>
> Thanks for your review. We will improve the paper with the requested changes.
>
> We will highlight in more detail the core of what makes the family of dynamics hard for black-box dynamics methods but easy for other methods. This should help recognize these families and maybe construct a formal characterization of them (which we do not know how to do).
>
> Here are the core properties that we integrate into our family of dynamics to make it hard for black-box dynamics methods:
>  - the dynamics have several actions to take, where each of them is critical to the success;
>  - each action can have a positive or negative effect depending on the dynamics;
>  - the dynamics provide non-informative feedback by aggregating the effects of all the actions into one reward, thus hiding the effects of each action to a method relying on reward feedback.
>
> But at the same time, the family is tractable to other methods because:
>  - a rewarding state can be inferred from random exploration.
>  - the effect of each action is revealed in the state-to-state transitions.

---

### Review · Reviewer_NLQh · 2024-12-02

**Summary Of Contributions:**

This paper presents a formalism that encompasses RL algorithms that use neural networks and a reply buffer. It then shows a class of MDP where a simple random exploration will fail to solve, while smarter exploration (in this case, Hindsight Experience Replay) will easily handle.

**Audience:**

Yes

**Broader Impact Concerns:**

No concerns.

**Claims And Evidence:**

No

**Requested Changes:**

I don't think simple changes would improve this paper.

My advice would be to take a larger look to the exploration in RL literature, and study what are the best existing methods to solving the introduced class of MDPs (not necessarily goal-based, like Direct and Diffuse or Max entropy approaches), compare them and eventually derive a new approach that works best on that particular class of problems.

**Strengths And Weaknesses:**

I'm am confused about the contribution of that paper.

It is possible that I missed a fundamental detail and completely misunderstood, but any MDP with a sparse reward will be easier to solve with a smart exploration-based approach. So I don't get the point of the introduced class of MDPs. And I don't get the point of the paper.

Another important confusion:
HER does respects the assumptions from def 3.2: just take f(s, D) := f(a | s, g), where f is trained just as described in Algo 15 (which is a gradient descent on a perceptron by the way). So according to Theorem 4.1, even HER should fail.

In practice it does not, because the assumption of the permutation used to prove Theorem 4.1 is actually incompatible with any classical neural network, which are position-dependent. But like HER, all RL algorithms (DQN, PPO, etc) use such networks and so don't respect the assumption 3.2. And the reason a simple PPO will fail is just because there is no exploration method implemented.

Finally, I found that paper particularly difficult to read, with a lot of weirdly composed sentences, and references to definitions / theorem before introducing them. A lot of introduced algorithms are extremely complexified to finally describe very simple process, like numeroting states for the encoder-decoder.

---

> ### Author Response · Authors · 2024-12-10
> **Answer**
>
> We thank the reviewer for his/her review.
> Below, we address the points raised sequentially:
>
>
> ---
> *"any MDP with a sparse reward will be easier to solve with a smart exploration-based approach. So I don't get the point of the introduced class of MDPs. And I don't get the point of the paper."*
>
> The point of the paper is to prove that a broad class of RL methods is limited in efficiency on some problems where other toy methods are not. Thus, our paper contributes to understanding the necessary algorithmic ideas for general-purpose efficient RL methods.
>
> It may seem intuitive that Hindsight Experience Replay (HER) could outperform our class of black-box dynamics RL methods (DQN, PPO, AlphaZero). However, a direct comparison is not fair since HER critically relies on a goal to be known a priori. Our paper goes beyond that unsatisfactory comparison by proving that no strong a priori knowledge is necessary to get an efficiency advantage over our class (Alg. 3, Thm 4.1). Additionally, our numerical results with a second toy method (Alg. 16) show that a planning procedure can also be sufficient to get this advantage.
>
> We note that we do not rely on advanced exploration schemes (like maximizing state diversity) to build our results. Alg. 3 only uses a uniform policy to sample trajectories and Alg. 16 additionally uses on-policy data. Moreover, maximizing state diversity would not help to solve the family of problems we define.
>
> ---
> *"HER does respects the assumptions from def 3.2: just take f(s, D) := f(a | s, g), where f is trained just as described in Algo 15 (which is a gradient descent on a perceptron by the way). "*
>
> We respectfully disagree, the HER method does not fit our Def 3.2. Yes, our definition allows to train neural networks as usually done by gradient descent, however, Def. 3.2 does not allow to train these neural networks using any data. In particular, Def 3.2 prohibits full access to future states in trajectories for training, this constraint is proved at the end of the proof of Theorem 4.1 in Appendix A. Since HER needs these future states to learn its state-to-state reachability function, HER cannot fit Def 3.2.
>
> We will add a note in the paper to explain this.
>
> ---
> *"the assumption of the permutation used to prove Theorem 4.1 is actually incompatible with any classical neural network, which are position-dependent. But like HER, all RL algorithms (DQN, PPO, etc) use such networks and so don't respect the assumption 3.2."*
>
> Def. 3.2 does not impose the learned neural networks to follow that constraint. Our Def 3.2 asks for a symmetry in the learning process but not in the learned neural networks, the learned neural networks can be position-dependent. More precisely, we ask for a symmetry between the dataset and the learned neural networks. We prove that this assumption holds for gradient-descent training in Appendix E.
>
> We will clarify this distinction in the paper.
>
> ---
> *"the reason a simple PPO will fail is just because there is no exploration method implemented."*
>
> As long as the exploration method fits Def. 3.2, we proved it will not help on our family of problems (Thm 4.1). If, however, an exploration method does not fit Def. 3.2 and helps solve the defined family of problems, it would be interesting to add it in our discussion section on RL methods avoiding the limitation.

---

### Author Response · Authors · 2024-12-16
**Revisions post-reviews**

We thank all the reviewers for their time and expertise given in the assessment of our work.

We updated the manuscript and highlighted all the additions in blue.
We summarize here these additions:
 - Adding a numerical experiment with a model-based method, we tested AlphaZero with its two players' components removed. (p.11; kyzM)
 - Adding more details/concrete example for the encoding process with the interface (p.3/4 and p.10; for UYVb)
 - Discussing generalization of the result toward more problems (p.10; UYVb)
 - Explaining differences with the proof of Sun and al. (2019). (p.17; UYVb)
 - To be explicit that the symmetry condition is on the learning process and not directly on the neural networks. (p.7; NLQh)
 - To be explicit that Alg. 3 is not black-box dynamics. (p.9; NLQh)

---

### Decision · Action_Editor_XH6a · 2025-01-26

**Recommendation:** Accept with minor revision

**Comment:**

The reviewers agree that the problem studied by this paper is important, did not find any major issue with the proofs and appreciated how the empirical results validate the theoretical findings. Some issues of clarity were already addressed by the authors.

The main issue that remains is on the scope of the class of RL algorithms studied in the paper (Definition 3.2.).
Improving the clarity of this point is crucial to convincingly support the claims of the paper, that are dependent on the scope of this definition.

I would like to stress that the minor revision I'm requesting is an important one: improving clarity, providing more examples, and thoroughly discussing the cases suggested by the reviewers is within reach, but fundamental to make the paper ready for publication.

In particular, I suggest to provide more intuition on why permutation invariance is so important.

**Audience:**

The results of this work would be of interest to the RL community.

**Claims And Evidence:**

Claims are substantiated. Results appear to be correct.

---

> ### Author Response · Authors · 2025-03-07
> **revisions**
>
> Many thanks for handling the review process!
>
> We updated the manuscript to its camera-ready version.
>
> Here is a summary of the changes made in addition to the previous revision.
>
>  - We discuss exploration techniques in or out of the studied class. [end of page 7; end of page 13]
>
> _These examples are archetypes of the main categories of RL methods and cover Q-learning with the Bellman equation, Policy Optimization with policy gradients combined with a critic, and black-box model predictive control with tree-based search. The last example also covers a mix of model-based tree-based search and model-free RL. We refer to the documentation of [Achiam 2018) for a taxonomy of RL methods. These examples also illustrate how several commonly used exploration methods fit our framework: $\epsilon$-greedy, on-policy, and optimism under uncertainty with UCB. Our proofs for these examples depict how other methods can be translated to fit our Definition 3.2._
>
> _We refer to [Ladosz 2022] for a survey on exploration in RL methods and follow their taxonomy. Our class of black-box dynamics methods covers some commonly used exploration techniques such as $\epsilon$-greedy and optimism under uncertainty with UCB, as shown in Appendix D. Other exploration techniques do not necessarily fit our class. Methods based on maximizing intrinsic rewards, state or behavior diversity, do not solely focus on the task-reward and are thus not covered by Definition 3.2. Also, methods based on exploratory goals might use universal value functions and Hindsight Experience Replay approaches which are also out of our class, as discussed above._
>
>  - We add a remark on why the permutation symmetry is important in the derivation our result [pages 3end/4start; page 10]
>
> _We ask $F$ to respect a constraint on its output without obstructing the implementation of our methods with the interface. Specifically, we demand that $F$ is invariant to shufflings of the coordinates in the states. This condition is naturally satisfied by ML algorithms since, without a priori, a learning process is expected to treat the features symmetrically. This condition allows us to model the reliance of methods on rewards to extract information from states. This is leveraged in our Theorem 4.1 to prove that some states are indistinguishable from the point of view of a black-box dynamics RL method._
>
> _In this analysis, the symmetry condition asked on $F$ is crucial. It allows us to prove that the outputs of $F$ are equal for different states in our problems. These states are thus indistinguishable, and imply that the algorithm fails to leverage the crucial information in them._
>
>  - We provide an example of the permutation symmetry condition applied to a linear regression. [page 7]
>
> _We stress that the symmetry restriction imposed on $F$ does not imply that the learned ML model is invariant to permutations of its input coordinates. Instead, the restriction imposes a symmetry between the learned ML model and the dataset used for its training. For example, when learning a linear model, if we permute the first two coordinates in the training data, then the resulting model will also have the values of its first two parameters permuted._